# H3.3 deposition counteracts the replication-dependent enrichment of H3.1 at chromocenters in embryonic stem cells

Stefano Arfè [1,3], Tina Karagyozova [1,5], Audrey Forest[1], Dominic Bingham [1], Hatem Hmidan[1,4], David Mazaud [1], Mickaël Garnier [1], Patricia Le Baccon [1], Eran Meshorer [2], Jean-Pierre Quivy [1] ✉ & Geneviève Almouzni [1] ✉

Chromocenters in mouse cells are membrane-less nuclear compartments representing typical heterochromatin stably maintained during cell cycle. We explore how histone H3 variants, replicative H3.1/2 or replacement H3.3, mark these domains during the cell cycle in mouse embryonic stem cells, neuronal precursor cells as well as immortalized 3T3 cells. We find a strong and distinct H3.1 enrichment at chromocenters, with variation in mouse embryonic stem cells. Mechanistically, this H3.1 selective enrichment depends on the DNA Synthesis Coupled deposition pathway operating in S phase challenged when we target H3.3 deposition through the DNA Synthesis Independent deposition pathway mediated by HIRA. Altering the H3.1/H3.3 dynamics at chromocenters in mouse embryonic stem cells affects nuclear morphology and cell division. Here, we reveal opposing mechanisms for H3.1 and H3.3 deposition with different enforcement according to cell cycle and potency which determine their ratio at chromocenters and are critical for genome stability and cell survival.

Chromatin states and their plasticity are associated with distinct cell fates[1]. Indeed, the versatility of the basic unit of chromatin, the nucleosomal core particle, can provide different flavors with the choice of histone variants and their histone post-translational modifications (PTMs)[2,3]. In mammals, two major non-centromeric histone H3 (H3.1/2 and H3.3) variants display distinct distribution patterns across the genome and use distinct deposition pathways. Notably, replicative variants, such as H3.1/2, are deposited via a DNA synthesis coupled (DSC) pathway promoted by the chaperone CAF-1 complex mainly during S phase and show a broad genome-wide distribution[4]. In contrast, H3.3, a replacement variant, is incorporated in a DNA synthesis-independent (DSI) fashion either by the chaperone HIRA at active chromatin regions and specialized nuclear domains[4,5] or by the DAXX-ATRX complex at constitutive heterochromatin regions, including telomeres, retrotransposons, and pericentric heterochromatin[6–9]. Heterochromatin is typically associated with high levels of constitutive heterochromatin marks, including H3K9me3 and HP1 proteins which disruption is often associated with increased DNA damage and defects in chromosome segregation[10,11]. In addition, heterochromatin integrity is also linked to chromatin replication since the perturbation of the replicative CAF-1 histone chaperone disrupts chromocenter formation in both ESCs and mouse embryos[12].

Chromocenters are membrane-less nuclear domains visible as DAPI-stained foci in interphase nuclei. In mouse cells, they correspond to the clustering of pericentric domains from different chromosomes, forming typical constitutive heterochromatin[13]. The pericentric domains, mostly composed of major satellite DNA repeats, flank the most centric region, which contains minor satellite DNA repeats and is enriched in CENP-A, the centromeric variant. These chromosomal landmarks play essential roles in genome organization and stability[13,14].

[1]Nuclear Dynamics, Institut Curie, PSL University, Sorbonne Université, CNRS, Paris, France. [2]Department of Genetics, The Alexander Silberman Institute for Life Science, and the Edmond and Lily Safra Center for Brain Sciences (ELSC), The Hebrew University of Jerusalem, Jerusalem, Israel. [3]Present address: Center for Neural Science and Medicine, Department of Biomedical Sciences, Cedars-Sinai Medical Center, Los Angeles, CA, USA. [4]Present address: Physiology and pharmacology department, college of medicine, Al-Quds University, Jerusalem, Palestine. [5]Present address: School of Biological Sciences, Institute of Cell Biology, University of Edinburgh, Edinburgh, UK. ✉e-mail: jean-pierre.quivy@curie.fr; Genevieve.almouzni@curie.fr

During development and cellular differentiation, chromocenters undergo dynamic remodeling, reflecting changes in chromatin and nuclear architecture[15–21]. The importance of histone variants in cell fate choices during development and disease[22] has further raised the importance of considering their relative distribution and its regulation. Indeed, the differential deposition of histone variants and distinct PTMs has emerged as an important player in establishing euchromatin and heterochromatin during development[15]. For example, facultative heterochromatin marks, such as H3K27me3, are critical for maintaining lineage-specific gene expression programs and H3.3 has been implicated in marking these regions[23,24]. While H3.3 exhibits a distinct genomic enrichment pattern, H3.1/2 variants are more broadly distributed and associated dynamically in the chromatin of totipotent and pluripotent cells[25,26]. Interestingly, in plants, chromocenters show a specific enrichment in H3.1[27]. In mouse embryonic stem cells (ESCs), during replication in heterochromatin regions, H3.1/2 recycling and maintenance is ensured both before and after differentiation, while active regions do not show such a stable maintenance[28,29]. Thus, understanding the mechanisms determining histone variant dynamics at distinct subnuclear regions, such as chromocenters can shed light onto means to achieve distinct genome marking in different cells.

Here, we investigate the nuclear distribution of H3.1 and H3.3 variants, focusing on their relative enrichment at chromocenters throughout the cell cycle and in different cellular states. Leveraging microscopy and chromatin immunoprecipitation followed by sequencing (ChIP-seq) analyses, we monitor the spatial and temporal dynamics of histone variants in pluripotent mouse ESCs, neuronal precursor cells (NPCs), and immortalized differentiated fibroblast cells (NIH-3T3). While we detect both histone variants in the whole nucleus, only the replicative variants H3.1/2 systematically stood out at chromocenters, regardless of the cell potency state. Mechanistically, we provide evidence that DSC deposition is the major driver for the replicative H3 variant accumulation at chromocenters while restricting H3.3. Moreover, by forcing H3.3 deposition via the HIRA DSI deposition pathway, we challenge this pattern and interfere with H3.1 accumulation at chromocenters. Under these conditions, we observe nuclear morphology alterations and cell division defects in mouse ESCs. We discuss how the distinct dynamics of deposition for the respective histone H3 variant can ensure proper chromatin organization with a significant impact on cell function.

## Results

### Mouse chromocenters show specific replicative H3 variant enrichment

To explore the subnuclear localization of the H3.1 and H3.3 variants in cells with different differentiation potentials, we introduced SNAP-HA-tagged H3.1 and H3.3 under the control of a Tet-ON system in mouse ESCs (Fig. S1a). We first verified that the Doxycycline-induced expression of exogenous H3.1-SNAP-HA and H3.3-SNAP-HA did not interfere with the pluripotency of these engineered ESCs as attested by OCT3/4 detection (Fig. S1b). We could readily visualize the nuclear H3.1-SNAP-HA and H3.3-SNAP-HA using SNAP-pulse labeling in vivo with TMR 18 h after the addition of doxycycline to the ESCs with no further increase in fluorescence signal beyond 48 h of treatment, indicating that the amounts of exogenously expressed H3.1 and H3.3 had reached equilibrium after 48 h of doxycycline treatment (Fig. S1c). Western blot analysis using total cell extracts further showed that, under these conditions, exogenous H3.1 or H3.3 histones represented less than 5% of the total histone pool (Fig. S1d), minimally interfering with the endogenous pool. We next monitored the incorporation of these exogenous H3.1 and H3.3 into chromatin by pre-extracting soluble histones before fixation[30,31] (Fig. 1a, b). By imaging, for H3.1 we observed two main patterns with either a marked enrichment at chromocenters ("enriched" pattern) or a diffuse distribution throughout the nucleus ("even") (Fig. 1b, upper panels 1 and 2). In

contrast, H3.3 showed either a weak depletion at chromocenters ("excluded") or a homogenously diffuse distribution throughout the nucleus ("even") (Fig. 1b, lower panels 3 and 4). Linescan quantification confirmed these patterns for the nuclear distribution of exogenous H3 variants (Fig. S1e). Importantly, using specific H3.1/2 and H3.3 antibodies in non-engineered SNAP cells (parental ESCs), we reproduced the same observations for endogenous H3.1/2 and H3.3 (Fig. 1c). Furthermore, we confirmed that H3.3 at chromocenters displayed "excluded" pattern in ESCs by using both confocal microscopy (Fig. S2a) and high-resolution microscopy with STORM and HiLO illumination, the latter allowing better optical sectioning for an improved contrast (Fig. 1d). We next examined the nuclear localization of the two H3 variants after differentiation. We differentiated ESCs into neural progenitor cells (NPCs) and generated NIH-3T3 cell lines constitutively expressing SNAP-HA-tagged H3.1 and H3.3. We used the loss of OCT3/4 detection as an indication of a loss of pluripotency of NPCs (Fig. S1b) and found identical expression levels for exogenous H3 variants in NPCs and 3T3 cells compared to ESCs (Fig. S1d). Furthermore, the localization of exogenous H3 variants in NPCs and 3T3 cells proved identical to ESCs, with a clear enrichment at chromocenters for H3.1 (along with HP1α) in contrast with H3.3 (Fig. S2b). Furthermore, we confirmed this subnuclear localization for endogenous H3.1/2 and H3.3 variants in all three cell types, ESCs, NPCs, and NIH-3T3 (Fig. 1c). We further quantified at chromocenters the immunofluorescence signal corresponding to the detection of endogenous H3.1/2 (and not H3.3) in pluripotent (ESCs) and differentiated cells (3T3). For this, we considered the ratio between signal intensity at chromocenters and the rest of the nucleus using a 3D-FIED (3-dimensional fluorescence intensity enrichment at domains) method[32]. The value for the ratio corresponding to H3.1/2 signal at chromocenters compared to the rest of the nucleus was consistently above 1, indicating a general enrichment, yet lower in ESCs compared to 3T3 cells (Fig. S2c, left panel). In contrast, the ratio for H3.3 reached around 1 indicating a homogenous distribution or even a weak depletion (Fig. S2c, right panel). Since we noticed some heterogeneity between cells with cases where cells did not display solely "enriched" (for H3.1/2) or "excluded" (for H3.3) but had also an "even" pattern (Fig. 1a, b), we quantified the proportion of cells with "enriched", "even", and "excluded" patterns in ESCs, NPCs, and 3T3 cells. We performed the analysis both on endogenous variants using H3.1/2 and H3.3 specific antibodies in cells that did not express exogenous histones (Figs. 1a and S1e) and in cells expressing exogenous SNAP-HA tagged H3 variants by TMR pulse labeling in vivo (Figs. 1c and S2b). We found that for endogenous H3.3, ~75% of cells displayed "even" and ~25% "exclusion" patterns at PHC for every cell line analyzed (ES, NPC, and 3T3) (Fig. 1e top panel). We obtained a similar distribution for exogenous H3.3 (Fig. 1e bottom panel). For H3.1, every cell line showed "enriched" and "even" patterns, with virtually no "excluded" pattern (Fig. 1e). However, in contrast to H3.3, H3.1 showed different proportions of each pattern in ESCs versus the differentiated cell types. While the differentiated cell lines (NPCs and 3T3) showed ~85% of "enriched" and ~15% of "even" patterns, the pluripotent ESC lines, instead, displayed reproducibly a lower frequency of cells with "enriched" patterns (~70%). Importantly, this difference between differentiated cells and ESCs was reproduced for both endogenous (Fig. 1e, top panel) and exogenously expressed H3.1 (Fig. 1e, bottom panel). These results are consistent with the increased enrichment of H3.1/2 at chromocenters in differentiated 3T3 cells when compared to pluripotent ESCs (Fig. S2c). They further reveal a less prominent H3.1/2 enrichment at chromocenters in ESCs compared to differentiated cells.

Next, in a genome-wide approach, we exploited the SNAP-HA-H3.1 and SNAP-HA-H3.3 tagged ESC lines and performed SNAP capture followed by high-throughput sequencing (SNAP capture-seq)[33,34] to map the specific H3.1 and H3.3 associated DNA sequences (Fig. S2d). We analyzed and quantified the distribution of H3.1 and

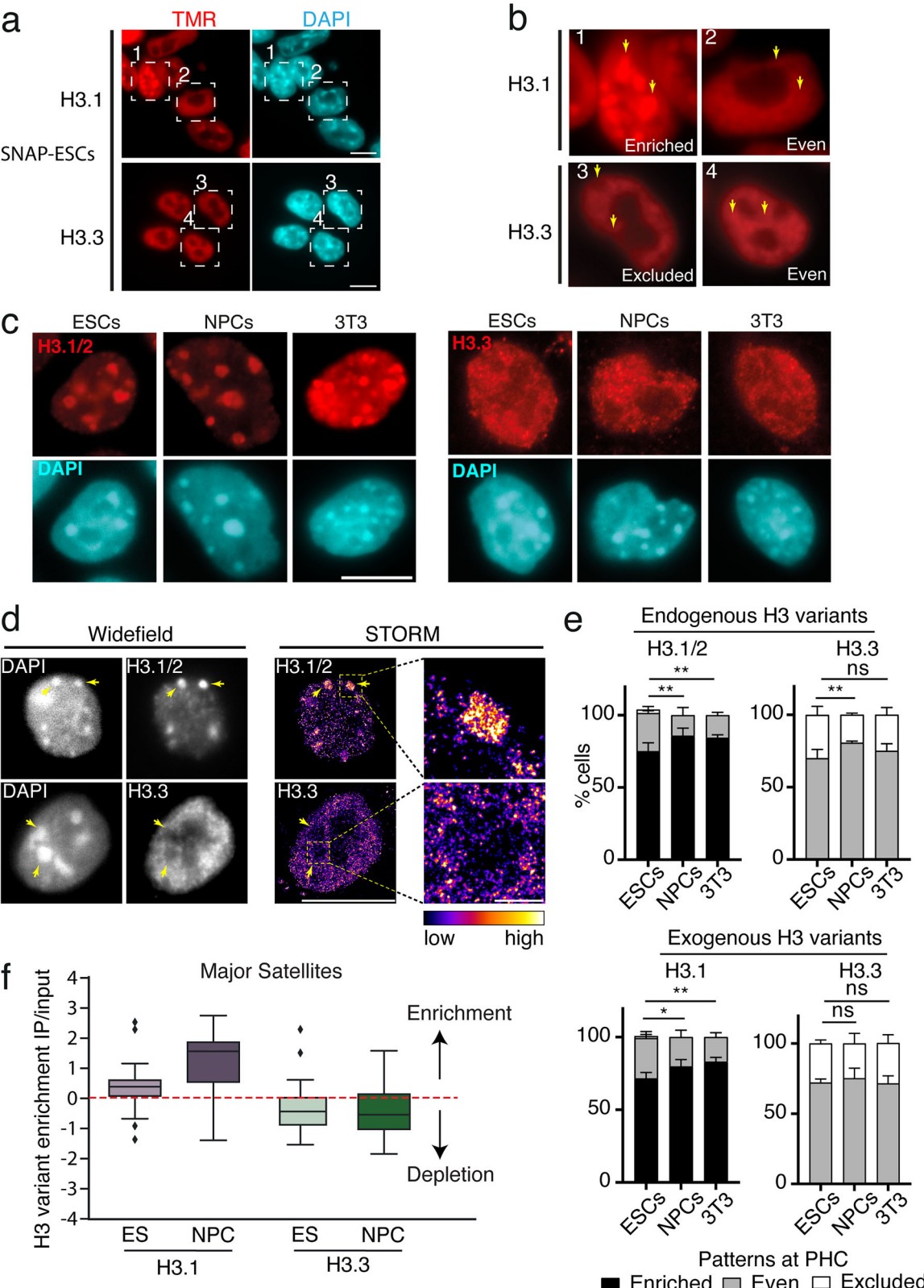

H3.3 at the Major satellite DNA repeats for both ESCs and NPCs. Importantly, we observed that during differentiation H3.1 enrichment increased steadily, with a relatively weak depletion of H3.3 at chromocenters (Fig. 1f). This is in line with our data using microscopy imaging. Thus, the genomic data confirm a specific H3.1 enrichment over chromocenter-associated repetitive elements, an enrichment that is maintained from pluripotent to differentiated cells. We

conclude that the specific subnuclear localization of histone H3 variants at chromocenters is a robust feature maintained during differentiation. However, we also note that interestingly, in differentiated cells, the H3.1 enrichment at the chromocenter is reinforced. Importantly, this analysis also underlines the distinct dynamics of enrichment for H3.1 and H3.3 at chromocenters during differentiation.

**Fig. 1 | Histone variant H3.1 is enriched at chromocenters in mouse cells while H3.3 is not. a** Representative wide-field epifluorescence images of Pulse-labeled (TMR) H3.1 and H3.3-SNAP (red) and DNA counterstaining with DAPI (cyan) in mouse ESCs. Single planes are shown, and squared boxes indicate nuclei on the right panels. Scale bar: 10 μm. **b** Insets of representative ESC nuclei showing "Enriched" (1), "Even" (2, 4), and Excluded (3) H3 patterns at chromocenters for H3.1/H3.3. Arrows point to clusters of pericentric chromatin (or chromocenters) identified as DAPI-dense regions. Scale bar: 10 μm. **c** Representative epifluorescent images of endogenous H3.1/2 (left) and H3.3 (right) detected with antibodies (red) in ESCs, neuronal progenitor cells (NPCs) derived from ESCs, and NIH-3T3 cells, followed by DNA counterstaining (DAPI, cyan). Scale bar: 10 μm. **d** Images from STORM acquisitions in HiLo illumination mode of endogenous H3.1 (top) and H3.3 (bottom) in ESC counterstained with DAPI. Arrows indicate chromocenters. HILO widefield mode, before STORM acquisition. (left) and super-resolved (right) images are shown. On the right panel, a zoom-in of the squared chromocenter is shown. The color scale indicates the density of localizations. Scale bar: 10 μm for nucleus,

1 μm for the zoom inset. **e** Quantitative analysis of the proportion (in %) of cells exhibiting recurrent patterns of endogenous and exogenous H3 variants at PHC in different cell backgrounds. Bar plots shows the mean and standard deviations (s.d.) of 100 nuclei for each cell line from 4 experiments for ESCs and 3 for NPCs and 3T3. ANOVA two-way test was used for statistical analysis: ns ($p > 0.05$), *($p < 0.05$), ** ($p < 0.01$). P values for endogenous H3 variants: H3.1/2 ESCs vs NPCs = 0.0013; H3.1/2 ESCs vs 3T3 = 0.0041; H3.3 ESCs vs NPCs = 0.0011; H3.3 ESCs vs 3T3 = 0.091. P values for exogenous H3 variants: H3.1 ESCs vs NPCs = 0.0010; H3.1 ESCs vs 3T3 = 0.0135; H3.3 ESCs vs NPCs = 0.619; H3.3 ESCs vs 3T3 = 0.9786. Source data are provided as a Source Data file. **f** Boxplot showing quantification of H3 variant enrichment at major satellite repeat elements in ESCs and NPCs by ChIP-Seq (SNAP-capture) from 1 replicate. The center of the boxplot is the median, the bounds of the box are the 1st and 3rd quartiles, diamonds are outliers, and the whiskers extend to $1.5 \times IQR$. The H3 variant enrichment is displayed as a Z-score of log2 enrichment of IP over input indicating enrichment when above 0 as indicated by red dotted line. Source data are provided as a Source Data file.

## H3.1 enrichment at chromocenters varies during cell cycle and potency

The fact that, in an asynchronous cell population, we did not detect in every cell a strong H3.1 enrichment at chromocenters suggested possible cell cycle variations. Thus, we monitored H3 patterns in parallel with cell cycle markers. We used Aurora B and EdU labeling, to discriminate the G1 (Aurora B−/EdU−), S (Aurora B-/EdU+), and G2 cells (Aurora B+/EdU−) (Fig. 2a). In addition, we used the spatiotemporal organization of DNA replication foci (based on EdU pattern) to follow S phase progression with (i) "Early S-phase" showing a high density of small foci throughout the nucleus; (ii) "Mid S-phase" with typical ring-shaped staining around the chromocenters; and (iii) "Late S-phase" with a few large foci at the nuclear periphery[35–37]. Additionally, within the S-phase cell population, we distinguished cells with chromocenters that had not yet replicated ("Early") from those undergoing replication ("Mid") and/or had already experienced replication ("Late"). First we found that H3.1 nuclear patterns changed with cell cycle progression (Fig. 2b, left panels) while H3.3 was stable (Fig. 2b, right panel). We estimated the proportion of cells displaying "enriched", "even", and "excluded" patterns at chromocenters for H3.1 and found that NPCs and 3T3 cells presented a high proportion (>80%) of "enriched" patterns at all cell cycle stages (G1, Mid S-phase, Late S-phase, and G2) except for Early S-phase (~50% "enriched") (Fig. 2c, top panels). Surprisingly, in ESCs, >80% of cells in Mid-S, Late-S, and G2 phases displayed an "enriched" pattern, like NPCs and 3T3 cells, while this "enriched" pattern dropped to 50% in G1. This is in sharp contrast with non-pluripotent NPCs and 3T3 cells which maintain the "enriched" pattern in G1 (~95%) (Fig. 2c). In addition, during early S-phase ESCs showed the lowest proportion of H3.1 "enriched" pattern (~10%) amongst all cell lines analyzed. When monitoring H3.3, a reciprocal picture emerged, namely cells with the highest H3.1 "enriched" pattern correlated with detection of cells showing H3.3 "excluded" pattern (Fig. 2c, H3.3 bottom panels). In contrast, when we observed the lowest H3.1 "enriched" pattern in G1 and Early S, most cells did not exhibit H3.3 "exclusion". (Fig. 2c, H3.3 bottom panels).

We then exploited a FUCCI mouse ESC model[38] to discriminate Early G1 (hCdt1++), Late G1/Early S (hCdt1+/Geminin+), Mid/Late S (hCdt1−/Geminin+), and G2 cells (Geminin++) (Fig. 2d). We monitored H3.1/2 patterns in parallel with cell cycle markers. In line with the above results, again we observed the lowest H3.1 "enriched" pattern in early G1 and early S phase (~5% and ~15%, respectively) followed by a strong increase of the same profiles during Mid/Late S and finally G2 (~80% and ~95%, respectively) (Fig. 2e). Taken together, these data further support the dynamic contribution at chromocenters for both H3.1/2 and H3.3. Furthermore, while differentiated cells show a steady enrichment of H3.1/2 enrichment at chromocenters throughout the cell cycle, during G1 ESCs exhibit a weaker H3.1 marking (less

"enriched" patterns), mirrored by H3.3 gain (less "excluded" patterns) and regained H3.1 enrichment following chromocenter replication.

## In S phase, DNA synthesis promotes H3.1 enrichment at chromocenters

To investigate key parameters leading to the enrichment of H3.1/2 (but not H3.3) at chromocenters, we considered the DNA synthesis coupled (DSC) and DNA synthesis independent (DSI) pathways promoting H3.1/2 and H3.3 deposition, respectively. H3 histone variants differ in their amino-acid sequence mainly in two regions: (i) in the histone fold domain: a motif that allows distinct recognition by histone chaperones; (ii) in the amino-terminal tail at position 31: a serine in H3.3 instead of an alanine in H3.1, which phosphorylation proved critical to activate transcription during key cell fate transitions and development[39–41]. The DNA synthesis coupled (DSC) deposition involves the CAF-1 complex and requires the SVM motif in H3.1/2[42,43]. For DNA synthesis independent (DSI) deposition associated with transcription[4,5,44], other histone chaperones interact with the AIG motif in H3.3[5,45–47] (Fig. 3a). Finally, H3.3 also differs from H3.1 by a serine to cysteine substitution at position 96, equally present in H3.2 (Fig. 3a). We generated transgenic ESCs lines with wild-type H3.1, H3.2, and H3.3 SNAP-tag constructs, along with an H3.1 construct with an A31S substitution in its N-terminus (A31S) and an H3.3 with either a phospho-mimic (S31D) or phospho-dead (S31A) substitution. We found that all constructs with the SVM motif (corresponding to DSC deposition) showed enrichment at the chromocenters (Fig. 3b, top). In contrast, constructs with the AIG motif (corresponding to DSI deposition), including the S31 phospho-mimic (S31D) and phospho-dead (S31A) mutants, did not accumulate at chromocenters (Fig. 3b bottom). Our quantification of the proportion of cells displaying "enriched", "even", and "excluded" patterns at chromocenters during S phase progression showed that constructs linked to DSC deposition led to "enriched" patterns (Fig. 3c). Importantly, chimera presenting either S31 (from H3.3) or S96 (residues only present in H3.2, Fig. 3a) showed no significant differences; thus, they did not interfere with the stable enrichment. In contrast, the DSI constructs did not show "enrichment" but led to "even" and "excluded" patterns during the S phase like endogenous H3.3 (compare Fig. 3c with Figs. 2c and S1b, ESCs). Notably, phospho-mimic (S31D) and phospho-dead (S31A) mutants led to similar proportions of "even" and "excluded" patterns compared to their WT counterparts.

Several H3 mutants, in cancers are referred to as "oncohistones"[48]. We monitored as above subnuclear organization and enrichment at chromocenters for the oncohistone mutations H3.3-K27M, H3.3-K27L, H3.3-G34R, and H3.3-G34V. We found that like H3.3, they did not show any enrichment patterns throughout the S-phase but the highest proportions of even and excluded patterns during early and late S,

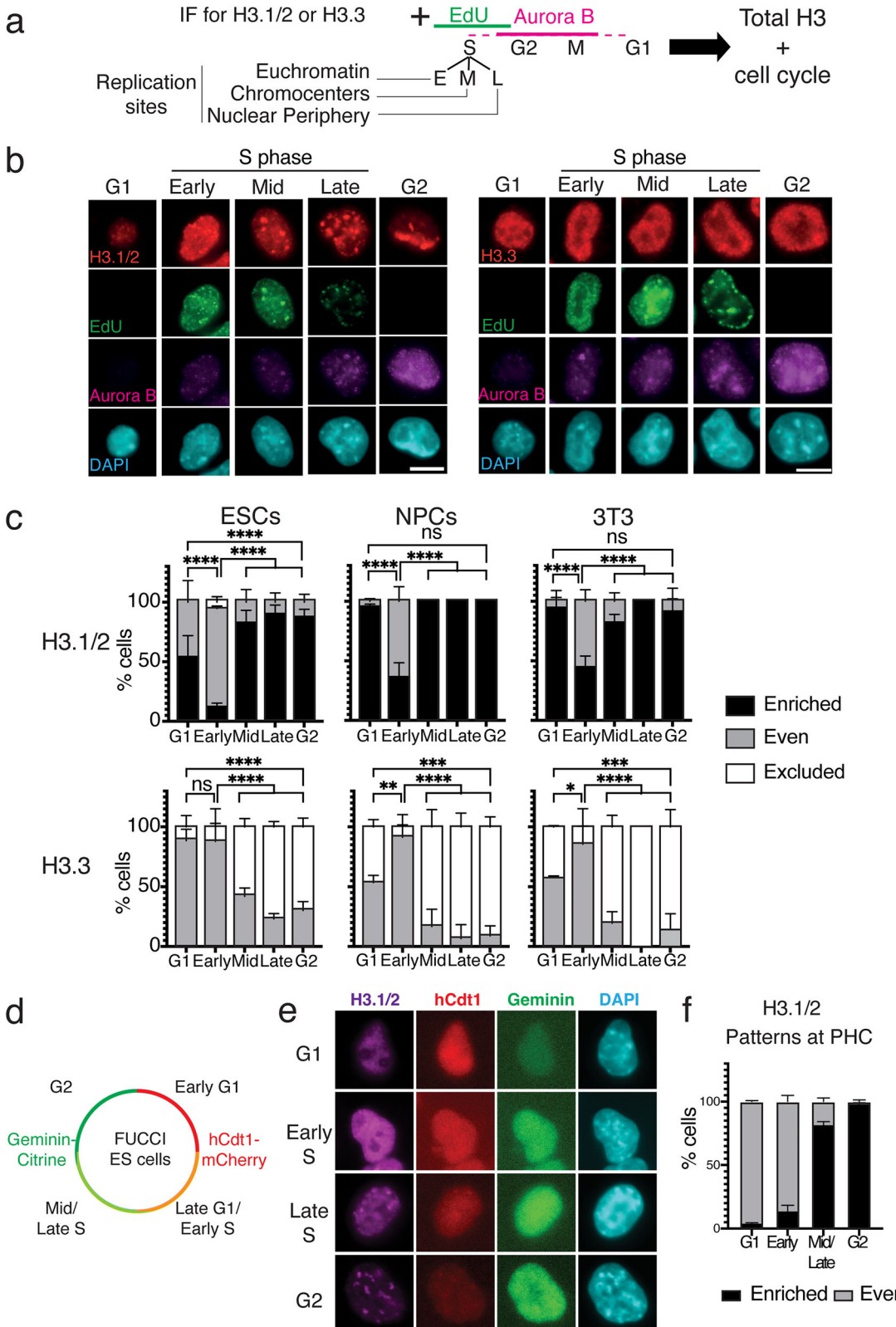

respectively, with little to no differences across the samples (Fig. S3). Notably, they followed the same dynamic changes of the DSI constructs, suggesting that these additional amino acid substitutions (K27 and G34) did not interfere with the dynamics of H3 deposition at chromocenters. The critical importance of the SVM motif enables us to conclude that a main parameter in defining H3.1 enrichment at chromocenters is the DNA synthesis-coupled deposition pathway.

## In Suv39 dn ESCs, H3.1 enrichment is compromised

We observed that ESCs displayed H3.1 loss (less "enriched" patterns) together with H3.3 gain (less H3.3 "excluded" patterns) (Fig. 2c) in G1 phase which is the exact time when Major satellite transcription increases[49]. This suggested to us that transcription at chromocenters could contribute in increasing H3.3-DSI mediated deposition and impact H3.1 enrichment. We used ESCs invalidated for both Suv39h1

**Fig. 2 | H3.1 enrichment at chromocenters follows chromocenter replication status. a** Experimental scheme for visualizing endogenous H3.1/2 or H3.3 during the S phase. Aurora B and EdU staining allowed to resolve the cell cycle stage of individual cells. Cells are scored as G1 (negative to Aurora B and EdU), S phase (EdU positive), or G2 (negative to EdU and positive to Aurora B). S-phase designations are based on EdU S phase patterns with the Early S phase (E) defined by a diffused staining (euchromatin); Mid S phase (M) by a focused EdU labeling around DAPI foci (chromocenters); and late S phase (L) by specific foci staining at the nuclear periphery. **b** Left: Representative immunofluorescence images of ESCs after in vivo labeling with EdU (green), immunofluorescence staining of H3.1/2 and H3.3 (red), Aurora B (magenta), and DNA (DAPI, cyan). Scale bars, 10 μm. **c** Quantitative analysis of the proportion of cells displaying H3 variant enrichment patterns at PHC in different cell lines (ESCs, NPCs, NIH-3T3) during the G1, S, and G2 phases. Stacked histogram show the mean (in %) and bar standard deviation from 4 experiments for H3.1 in ESCs and 3 experiments for the other. For H3.1, $n$ = 523, 329 and 275 nuclei

for ESCs, NPCs, and 3T3, respectively. For H3.3 $n$ = 328, 298, and 266 nuclei for ESCs, NPCs and 3T3 respectively. ANOVA two way test was used for statistical analysis: ns ($p > 0.05$), *($p < 0.05$), **($p < 0.01$). $P$ values are shown in the source data file. Source data are provided as a Source Data file. **d** Schematic of FUCCI ESCs allowing visualization by microscopy of hCdt1-mCherry and Geminin-Citrine during the cell cycle. Cells are scored as G1 (mCherry++/Citrine−), Late G1/Early S (mCherry+/Citrine+), Mid/Late S (mCherry−/Citrine+), or G2 (mCherry−/Citrine++). **e** Representative immunofluorescence images of FUCCI ESCs expressing hCdt1 (red) or Geminin (green) with immunofluorescent staining of H3.1/2 (magenta) in G1, Early S, Late S, and G2 phases. Scale bars, 10 μm. **f** Quantitative analysis of the percentage of FUCCI ESCs exhibiting recurrent H3.1/2 patterns at chromocenters throughout the cell cycle. Bar plots show the mean and standard deviation of 100 nuclei for each cell cycle phase from 4 experiments. Source data are provided as a Source Data file.

and Suv39h2 (Suv39h dn)[50] in which the transcription of MajSat was found increased[51,52]. In Suv39h dn, we found a decrease in H3.1 enrichment at chromocenters when compared to wt ESCs (Fig. 3d). Taken together, our results show that the SVM motif (DSC pathway) defines H3.1/2 enrichment at chromocenters. They also raise the possibility that the H3.3 deposition DSI pathway linked to transcription to interfere and challenge H3.1 enrichment.

### Targeting HIRA at chromocenters out-competes H3.1 enrichment in ESCs

Since we observed that H3.1 enrichment patterns were mirrored by H3.3 depletion (Figs. 1e, f, 2c, 3d, S2c), we wondered whether forcing H3.3 deposition at chromocenter could interfere with H3.1 enrichment. To test this hypothesis, we engineered a transcription activator-like effector (TALE) designed to bind specifically to the major satellite DNA repeats[53,54] in fusion with the H3.3 chaperone HIRA tagged with a Clover fluorescent protein (Fig. 4a). Importantly, for this "forced" deposition of H3.3, we chose HIRA, a chaperone that do not "normally act at the site" in a WT background, rather than overexpressing a histone chaperone already present at PHC such as DAXX/ATRX. The TALE-HIRA fusion fulfilled these requirements to challenge a situation and perturb chromatin dynamics (in this case, H3.1/H3.3 enrichment), a key prerequisite to further develop our imaging analysis pipeline.

We used HIRA wild type and the mutant HIRA W700A-D800A unable to interact with CABIN1 and thus unable to promote de novo H3.3 deposition[55] and the TALE fused to Clover as control (Fig. 4a). For simplicity, we named TALE-Clover, Ctr; TALE-HIRA wt-Clover, HIRA wt; and TALE-HIRA W700A-D800A-Clover, HIRA mut. We next transfected ESCs with these constructs and visualized endogenous H3.1/2 and H3.3 by immunofluorescence both in the cells expressing the TALE constructs (Clover positive) and in those not expressing the TALE constructs (non-transfected NT, Clover negative). We first verified that HIRA wt and HIRA mut readily localized at the DAPI dense chromocenters as the Ctr[54], indicating that HIRA fusion proteins did not affect the TALE-mediated specific targeting at chromocenters (Figs. 4a and S4). By confocal microscopy, only the HIRA wt construct gave rise to H3.3 colocalizing with H3.1 at chromocenters, but not the controls (non-transfected clover negative cells (NT) and the constructs Ctr and HIRA mut were unable to promote H3.3 deposition) (Fig. S4). We next investigated whether the presence of TALE-HIRA impacts H3.1/2 and H3.3 at chromocenters. We observed that cells expressing HIRA wt had a global H3.3 increase in the nucleus and less exclusion of H3.3 localization at chromocenters in contrast to control cells (NT, Ctr, and HIRA mut expressing cells) (Fig. 4a, right). Most remarkably, we noticed a reduction in the H3.1 signal at chromocenters, which was not observed for the HIRA mut, the mutant deficient for H3.3 deposition (Fig. 4a right). Next, we quantified the proportion of cells displaying "enriched", "even", and "excluded" patterns for H3.3 and H3.1/2 in cells with TALE fusions at chromocenters (Clover positive cells) (Fig. 4b).

For H3.3, HIRA wt, but not TALE mut, increased the proportion of cells with the "even" pattern with a concomitant decrease of cells with the "excluded" pattern (Fig. 4b, left). Remarkably, HIRA wt led to a decrease in the proportion of cells displaying H3.1 enrichment at chromocenters (77–43%, Fig. 4b right). To confirm this effect, we quantified by 3D-FIED macro the relative enrichment of H3.1/2 and H3.3 at chromocenters targeted by HIRA wt and HIRA mut. We analyzed the cells expressing HIRA wt or HIRA mut (Clover positive cells) and used the non-transfected Clover negative cells (NT) as control (Fig. 4c). We found for H3.3 nearly identical enrichment for every condition (Fig. 4d, top panel). We detected a small but significant increase of H3.3 enrichment when comparing negative and positive cells, for both TALE wt and HIRA mut. However, we did not detect any significant difference in H3.3 enrichment between wt and HIRA mut when we compared positive cells only, indicating that the increased H3.3 enrichment at chromocenters visualized by immunofluorescence (Figs. 4a and S4) cannot be appreciated using this approach (Fig. 4c, left). We next performed the same quantification for H3.1/2 and found a significant decrease of H3.1 enrichment specifically in cells expressing HIRA wt when compared to HIRA mut and NT control cells (Fig. 4c, right). Taken together, these data indicate that targeting HIRA at chromocenters leads to a small but detectable enrichment of H3.3 at chromocenter concomitantly with a decrease of H3.1/2. Given that increased transcription at chromocenter in Suv39 dn ESCs is associated with decreased H3.1, enrichment (Fig. 3d), we also wondered whether the decrease in H3.1 enrichment could reflect transcription when HIRA wt was tethered at chromocenter increases H3.3. We used RNA FISH to quantify major satellite transcripts foci selectively in TALE constructs expressing cells (Fig S5a). We did not find differences in the proportion of cells displaying foci (Fig. S5b), nor in the amounts of foci per nucleus (Fig. S5c) between the 3 TALE constructs. We next wondered if alterations in H3 histone variants balance at chromocenters could impact the centric regions located next to the pericentric regions. We found that the nuclear localization of CENP-A (the histone H3 variant specifically enriched at centric domains) remained identical between Ctr, HIRA wt, and HIRA mut (Fig. S5c). These data indicate that, within the limits of our experimental design, targeting HIRA at chromocenters alters the dynamics of H3.1/2 and H3.3 deposition at the pericentric regions of the centromere without altering the transcriptional status of the chromocenter and the organization of the centric regions. Based on these data we propose that the targeted HIRA-mediated H3.3 deposition to chromocenters, on its own, acts as a dominant mechanism to out-compete H3.1 presence.

### HIRA at chromocenters impacts nuclear shape and cell division

The question that arises is whether impairing H3.1 enrichment at chromocenters affects its function in ESCs and has any impact on cell cycle and mitosis. We thus examined whether targeted H3.3-deposition at chromocenters affects constitutive heterochromatin hallmarks,

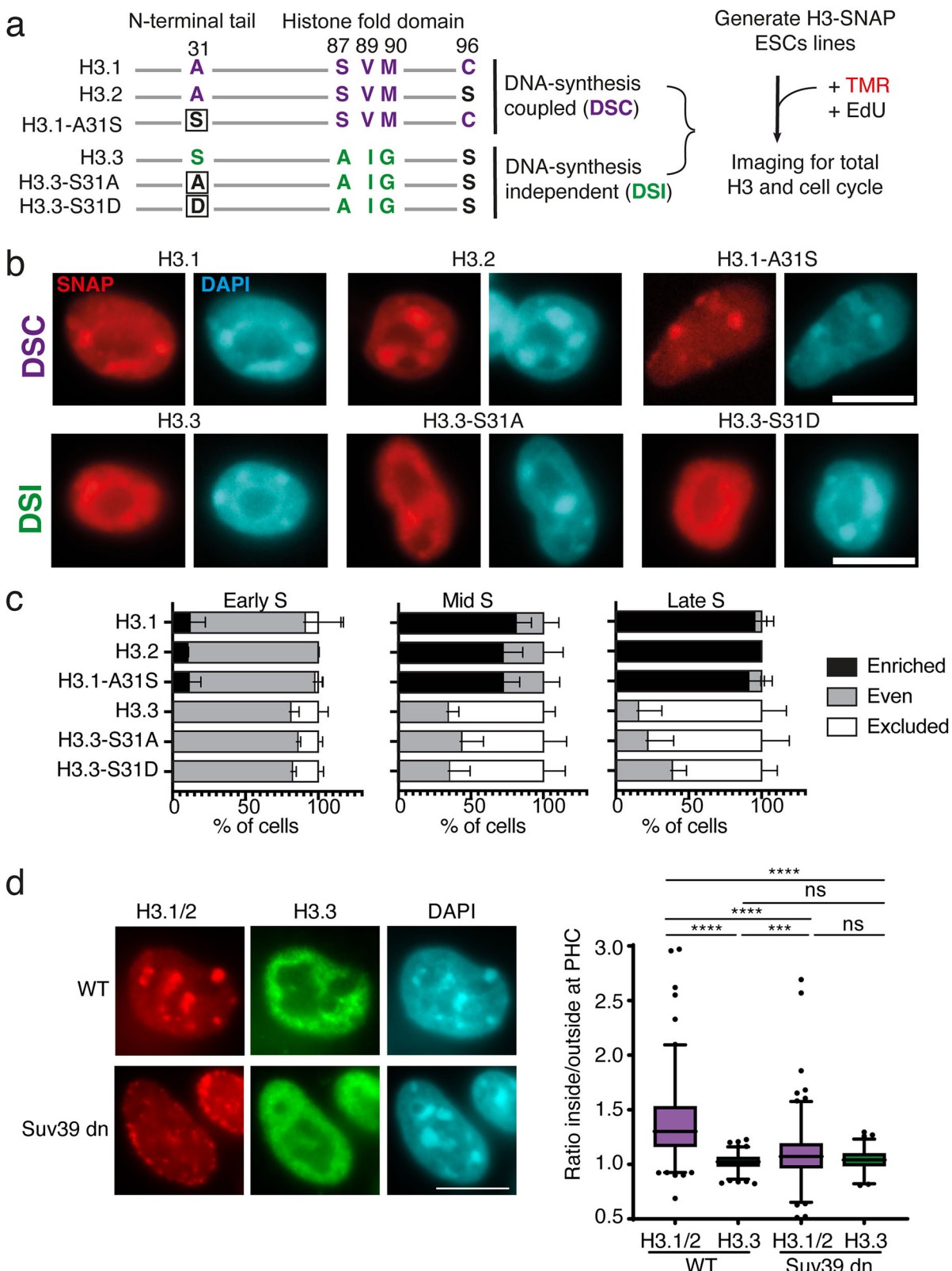

including HP1α and H3K9me3, and impacts cell viability during cell division. We monitored by immunofluorescence H3K9me3 and HP1α at chromocenters in cells targeted by TALE constructs (Fig. 5a, b left). We found an enrichment for both H3K9me3 and HP1α at chromocenters in Ctr, HIRA wt and HIRA mut (Fig. 5a, b left). By quantifying this enrichment at chromocenters by 3D-FIED, we did not detect significant changes for H3K9me3 enrichment (Fig. 5a left). However,

we observed a significant enrichment in HP1α for HIRA wt positive cells when compared to control cells (clover negative non-transfected cells (NT) and HIRA mut cells) (Fig. 5a right). While the different behavior between HP1α and H3K9me3 may be surprising, it is interesting to consider that an increase in HP1α may be independent of the modification, possibly linked to other binding properties of HP1α like DNA or RNA for example. Nevertheless, these observations prompted us to

**Fig. 3 | H3.1 enrichment at chromocenters requires SVM motif and is compromised in Suv39h dn ESCs. a** Left: Specific aminoacid residues are depicted for H3.1/2 (purple) and H3.3 (green) and grouped based on their similarities with the histone chaperone recognition motif (SVM or AIG). Aminoacid substitutions from original sequences are boxed. Right: experimental scheme **b** Representative epifluorescence images of H3-SNAP-Tag histones (red) in ESCs along with DNA counterstaining (DAPI, cyan). Scale bar 10 μm. **c** Quantification of cells exhibiting H3 patterns at PHC during Early, Mid, and Late S stages. Stacked histograms show the mean (in %) and bars s.d. from 3 experiments for all constructs exept H3.2 with 2 experiments. *n* = 115 for H3.1; 297 for H3.3; 120 for H3.2; 351 for H3.1 A31S; 186 for H3.3 S31A; 182 for H3.3 S31D. Source data are provided as a Source Data file. **d** left: Representative epifluorescence images of endogenous H3.1/H3.1 (Red) and H3.3 (Green) co-stained in WT and Suv39h dn ESCs along with DNA counterstaining

(DAPI, cyan). Scale bar, 10 μm. Right: boxplots showing the ratio of H3.1/2 (magenta) and H3.3 (green) immunofluorescence signal inside and outside of pericentric heterochromatin (PHC) in WT and Suv39h dn cells from 4 experiments. The center of the boxplot is the median, the bounds of the box are the 1st and 3rd quartiles, dots are outliers, and the whiskers extend to 1.5x IQR. For ESC WT n = 244 for H3.1 and 218 for H3.3; for Suv39h dn ESCs n = 294 for H3.1 and 155 for H3.3. Mann–Whitney two-tailed was used: **** = *p* value < 0.0001, *** = *p* value < 0.001, ns non-significant = p value > 0.05. P values: H3.1 ES WT vs. H3.3 ES WT < 1.e$^{-15}$; H3.1 ES WT vs. H3.1 Suv39dn < e$^{-15}$; H3.1 ES WT vs. H3.3 Suv39dn < e$^{-15}$; H3.3 ES WT vs. H3.1 Suv39dn = 0.00022247; H3.3 ES WT vs. H3.3 Suv39dn = 0.68257230; H3.1 in Suv39dn vs. H3.3 in Suv39dn = 0.2479430. Source data are provided as a Source Data file.

consider whether HIRA targeting and the changes in H3.1/2 vs H3.3 at pericentric heterochromatin could possibly affect centromeric function during cell division. We tested this hypothesis, by monitoring with time-lapse microscopy the time ESCs spent during cell division and the frequency of cell death events after mitosis. We transfected Ctr, HIRA wt, and HIRA mut constructs in ESCs and followed cell division events for 24 h corresponding to at least 1–2 cell divisions (Fig. 5c). We determined the cell division time as the time spent from prophase, when cell rounds-up (1 cell) to the end of telophase when the two daughter cells re-flatten onto the plate (2 cells) (Fig. 5c, yellow box, 5d). In our hands, cells expressing Ctr and HIRA mut showed an average cell division time of 45 min, whereas for HIRA wt expressing cells, the average time increased up to 75 min (Fig. 5d). This prolonged time spent in mitosis is accompanied by an increased loss of cells due to a fraction of cells that after rounding-up do not divide anymore and finally die (Fig. 5e). We also monitored nuclear morphology in cells expressing the TALE constructs by microscopy on fixed cells. In HIRA wt expressing cells, we detected an increased proportion of nuclei with altered morphologies, including heterogeneous shape, bigger size, multilobed when compared to nuclei in cells expressing the Ctr and HIRA mut constructs (Fig S6). Importantly, we did not observe major changes in cell division time (Fig. 5d), proportion of lost cells (Fig. 5e), and altered nuclei (Fig. S6b) between ESCs expressing Ctr and HIRA mut. This indicates that targeting the HIRA protein lacking its ability to promote H3.3 deposition at the chromocenter, does not lead to major perturbations. Based on these results using the HIRA targeted DSI deposition at chromocenters, we conclude that it is by interfering with H3.1 enrichment that defects arise in cell division impacting cell viability.

## Discussion

By exploiting a combination of imaging and sequencing methods, we defined how chromocenters show a controlled enrichment of the replicative H3.1/2 rather than the replacement H3.3 histone variant with an important function for cell division and survival. First, we found that chromocenters irrespective of the cell differentiation potential, show a reproducible H3.1/2 enrichment paralleled by a relative H3.3 depletion. This enrichment for replicative histone variants depends on the deposition pathway mediated by the CAF-1 histone chaperone in S phase (Fig. 6a). It is robustly maintained throughout the cell cycle in differentiated cells compared to pluripotent ESCs. In the latter, during the G1 phase, a loss of H3.1 enrichment occurs at the expense of a gain in H3.3 deposition possibly as a consequence of increased transcription at chromocenters (Fig. 6). We experimentally challenged the H3.1 enrichment at chromocenters by forcing H3.3 deposition through the artificial targeting of HIRA to chromocenters (Fig. 6). These conditions compromised heterochromatin marks at chromocenters, and cells showed defects in nuclear morphology and cell cycle. Whether artificial targeting using the ATRX/DAXX complex might result in a different outcome would be interesting to explore especially considering its ability to recruit

H3K9me3/HP1α to assess whether it could promote reduced H3.1 deposition and changes in nuclear morphology. At this point in time, our data thus reveal a conserved function for the DSC-deposition pathway to sustain H3.1 enrichment at chromocenters and to restrict the DSI-deposition machinery presence at pericentric heterochromatin domains during the cell cycle (Fig. 6). The changes in cell cycle following differentiation offers different opportunities for the respective contributions of the DSC and DSI deposition pathways.

In mouse cells, the general view posits that replacement variant H3.3 is enriched at active sites, enhancers, and promoters, but H3.3 also marks heterochromatin close to telomeres and pericentric regions[5]. However, as shown in our study H3.1 is also present in heterochromatin regions, visibly detected when discernable compartments form in the nuclear space. Notably, in plants, H3.1 is enriched in chromocenters, including both the centromeric and pericentromeric chromatin regions[27,56–58]. This series of observations raise key questions as to how such choices in histone variants with distinct dynamics can be achieved in nuclear domains. How does the observation in plants relate to mammals? How can the distinct dynamics of these variants explain changes associated with cell states during the cell cycle and differentiation?

We focused on chromocenters to follow the subnuclear distribution of H3.1 and H3.3 in pluripotent ESCs, NPCs, and terminally differentiated cells NIH-3T3 cells. As found in plants[27], we detected a consistent H3.1 enrichment at chromocenters and relative H3.3 exclusion. Genome-wide and image analysis confirmed this dual presence of H3.1 and H3.3 at major satellites and close to centromeres as previously reported in human cells and plants[59,60]. We further dissected by amino acid substitution key residues present in H3.1/2, but not in H3.3, which proved crucial for H3.1 enrichment at chromocenters. This analysis revealed the major importance of the histone fold domain of the SVM motif, which is key for the deposition involving the CAF-1 histone chaperone. Thus, we established that the major mechanism involved in replicative H3 enrichment is strictly dependent on the DNA synthesis deposition pathway. H3.1 enrichment at chromocenters varied during the cell cycle in sharp contrast with the steady H3.3 presence (Fig. 2). The relative decrease in the early S-phase in H3.1 at chromocenters can be explained by the global incorporation of H3.1 during early replication of euchromatin while heterochromatin in chromocenters only replicates later in mid-late S phase (Fig. 2b, left panel, Early S-phase). In agreement with this interpretation, this relative decrease is compensated during mid-S and late-S when chromocenters replicate and heterochromatin doubles, and this status is maintained in subsequent G2 and propagated in G1. This observation leads us to consider that H3.1 enrichment depends first on its deposition during S phase, but also on the possible competing effect due to deposition of H3.3 that could kick out H3.1. For every cell line analyzed, except for ESCs, the early-S phase showed few cells with H3.1 enrichment, and the peak of H3.1 enrichment consistently occurred during the mid-S, late-S, and G2/G1 transition phases. This is in line with our interpretation since

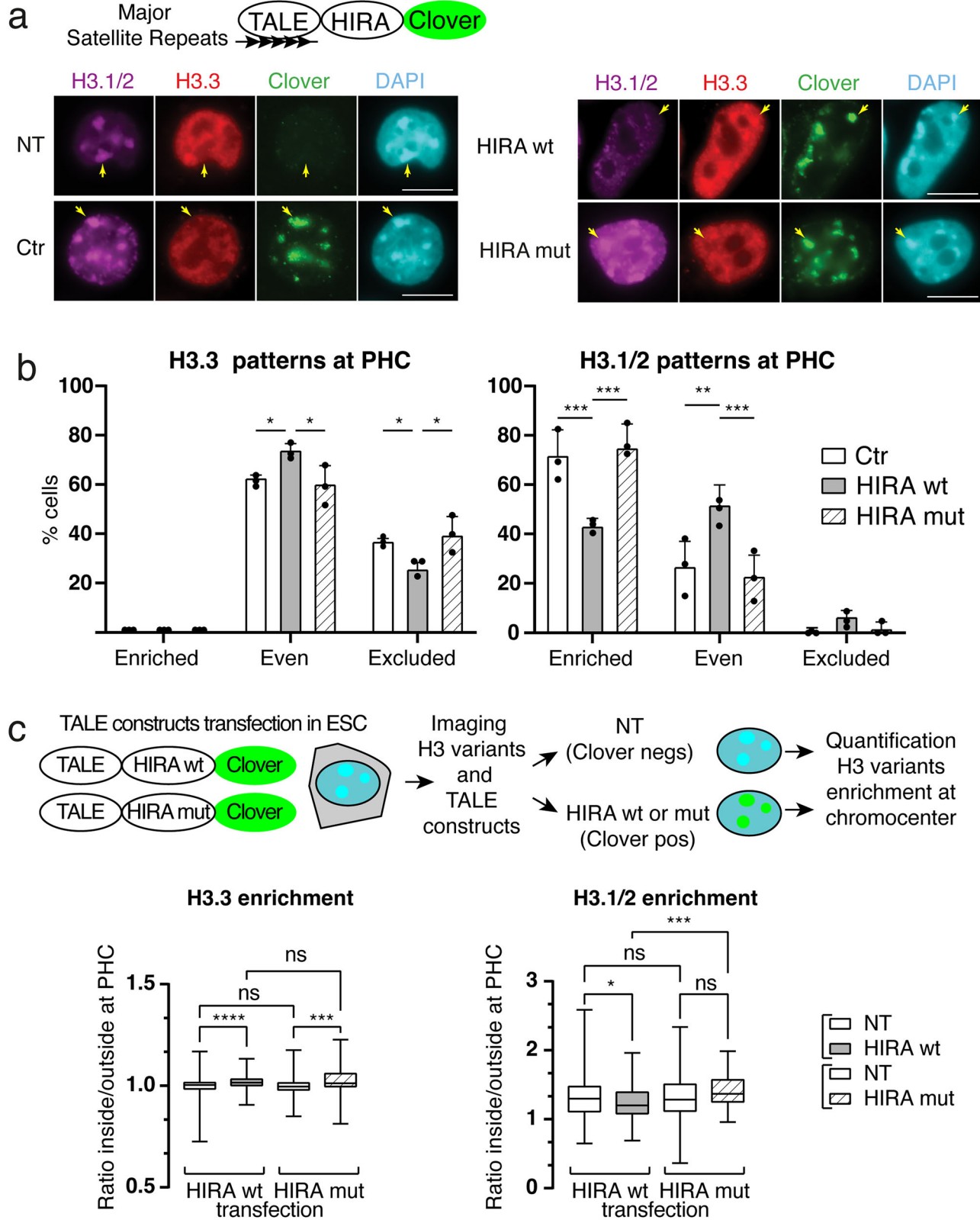

euchromatin regions replicate during the early-S phase and increased amounts of H3.1 are expected due to replication-coupled deposition, thereby leading to the observed relative decrease in H3.1 enrichment at chromocenters which replicate later. Indeed, when chromocenters replicate in the mid-S and late S phase, the relative enrichment between chromocenters and the rest of the nucleus is restored. We found the same behavior for these dynamics of H3.1

relative enrichment in NPCs and NIH-3T3 cells. However, the situation is different in ESCs, which transcribe major satellite repeats at higher levels when compared to differentiated cells[61]. Indeed, this is in line with previous reports showing that preventing major satellite transcription in ESCs enables the formation of stable chromocenters[62]. Thus, the unusual dynamics for replicative H3 observed in ESCs could ensure a stable heterochromatin

**Fig. 4 | Targeted H3.3 deposition via the HIRA complex alters local H3.1 enrichment at chromocenters. a** Top: scheme depicting HIRA Clover chimeric protein fused to TALE protein recognizing Major Satellite repeat DNA sequences. Bottom: representative immunofluorescent images of ESCs non transfected with TALE construct (NT) and expressing TALE constructs Ctr, HIRA wt and HIRA mut. Clover (green), endogenous H3.1/2 (magenta), H3.3 (red), and DNA (DAPI, cyan) are shown. Arrows point to chromocenter. Scale bars, 10 μm. **b** Quantitative analysis of the percentage of cells expressing Ctr, HIRA wt, and HIRA mut displaying "enriched", "even", and "excluded". Bar plot shows the mean and bars s.d. from 3 experiments. 100 nuclei were counted per condition. Dots indicate mean of each experiment. Source data are provided as a Source Data file. ANOVA two way test was used for statistical analysis: ns (p > 0.05), *($p$ < 0.05), **($p$ < 0.01). P values for H3.3 patterns at PHC: even Ctr vs HIRA wt = 0.0455; even HIRA wt vs HIRA mut = 0.0152; excluded Ctr vs HIRA wt = 0.045; excluded HIRA wt vs HIRA mut = 0.0152. P values for H3.1/2 patterns at PHC: enriched Ctr vs HIRA wt = 0.0006; even HIRA wt vs HIRA mut = 0.0002; even Ctr vs HIRA mut = 0.0021; even HIRA wt vs HIRA mut = 0.0005. Source data are provided as a Source Data file. **c** Top: scheme depicting approach to select cells non-transfected (NT) non expressing TALE

construct from cells expressing HIRA wt or HIRA mut for quantification of H3 variant enrichment at chromocenter. Bottom: boxplots of enrichment values of H3.3 and H3.1/2 at chromocenters in ESCs expressing HIRA wt or HIRA mut and nontransfected (NT) ESCs. The center of the boxplot is the median, the bounds of the box are the 1st and 3rd quartiles and the whiskers extend to max and min. For each HIRA wt and HIRA mut transfection, non-transfected cells (NT non-expressing TALE construct) and cells expressing HIRA wt or HIRA mut are indicated. Data are from three experiments. For H3.3 HIRA wt transfection $n$ = 206 and 109 for NT and HIRA wt; for H3.3 HIRA mut transfection, $n$ = 191 for NT and 41 for HIRA mut; for H3.1/2 HIRA wt transfection, $n$ = 227 for NT and 96 for HIRA wt, for H3.1/2 HIRA mut transfection, $n$ = 161 for NT and 50 for HIRA mut. Kruskal-Wallis test was used for statistical analysis: ns (p > 0.05), *($p$ < 0.05), ***($p$ < 0.001), **** ($p$ < 0.0001). $P$ values for H3.3 enrichment NT(wt) vs HIRA wt = 0.0000121; NT(wt) vs NT(mut) = 0.2957; HIRA wt vs HIRA mut = 0.95; NT(mut) vs HIRA mut = 0.0002. $P$ values for H3.1/2 enrichment NT(wt) vs HIRA wt = 0.0207; NT(wt) vs NT(mut) = 0.0920; HIRA wt vs HIRA mut = 0.0004; NT(mut) vs HIRA mut = 0.5713. Source data are provided as a Source Data file.

maintenance while preserving pluripotency ability[25,63,64]. Furthermore, the dynamics of structural chromatin proteins, including replicative histone variants and HP1, are constantly challenged and need to be replenished in stem cells[25,26]. Importantly, in Suv39 dn ESCs, we found that H3.1 enrichment is compromised (Fig. 3d), in line with the higher transcription reported in pericentric heterochromatin[51,52]. Considering that forced HIRA-mediated H3.3 deposition challenged H3.1 deposition at the chromocenter, it is possible that the increased major satellite transcription allows more H3.3 deposition. Thus, it will be important to examine further how the enrichment of labile H3.1 and the turnover of H3.1/H3.3 at the chromocenter in ESCs relates to transcription and its connection with mechanisms of histone deposition. Remarkably, when forcing H3.3 deposition, we found that chromocenters with decreased H3.1 enrichment showed severe defects in nuclear morphology and cell division (Fig. 6). Intriguingly, we monitored a modest increase in HP1α enrichment in these conditions. In this respect, it is interesting to note that increased HP1α at chromocenter following depletion of Major satellite transcripts has also been associated with chromosome instability and defective mitosis[62]. Future work should aim at dissecting in more details the intricacy of the relationships between the nature of H3 variant, transcriptional activity, HP1α and cell division.

While the normal setting of DSC and DSI favors replicative histones, it can be challenged and vary according to the cell potential. The question that arises then is whether the more "labile" H3.1 enrichment at chromocenters in ESCs simply reflects a generally higher H3.1/H3.3 turn-over or if it could have a more direct role specifically at chromocenters. Indeed, chromocenters are less well-defined in size and shape in ESCs than in differentiated cells. Interestingly, knocking down the p150 subunit of CAF-1, which impairs H3.1 deposition, leads to a loss of chromocenters in ESCs, but not in differentiated MEF cells[12]. These contrasting data underline the stronger dependency on the replicative deposition pathway in ESCs, while additional maintenance mechanisms come in place in differentiated cells that will have to be identified.

Importantly we found that replicative H3.1/2 and the replacement H3.3 histone variants can both be present at pericentric heterochromatin domains. However, their relative accumulation is a matter of dosage controlled by the replicative H3.1 deposition mechanism as opposed to replacement H3.3 deposition. This histone enrichment at chromocenters depends on the cell type and their cell cycle properties as summarized in our schematic model (Fig. 6). Therefore, it is tempting to consider that regulating the histone chaperones involved in distinct deposition pathways could represent an attractive means for shaping nucleosomal composition at distinct nuclear domains. Exploring these issues represents exciting avenues to better understand the control in nuclear organization and plasticity during cell fate decisions.

## Methods
### Cell culture
We cultured all cells at 37 °C in 5% CO2: KH2 mESCs[65] and Suv39 dn mESCs [50]on gelatinized feeder-free tissue culture plates in ESC media (Dulbecco's modified Eagle's medium supplemented with GlutaMax, Pyruvate and 4.5 g/L D-Glucose (Thermo Fisher Scientific), 15% calf fetal serum (Eurobio), 1000 U/ml penicillin/streptomycin, 1× MEM non-essential amino acids, 125 μm beta-mercaptoethanol supplemented with 1000 U/ml LIF (Millipore) and 2i inhibitors, which include 1 μM MEK1/2 inhibitor (PD0325901) and 3 μM GSK3 inhibitor (CHIR99021). To generate NPCs, we cultured mESCs cells in NPC media (ESC media (without LIF and 2i) supplemented with 1 μM Retinoic Acid (RA), 1× N-2 Supplement (Gibco) and 1× B27 Supplement (Gibco) for 4 days, and 3 additional days in the same NPC media supplemented with 10 ng/ml of FGF (Peprotech) and 20 ng/ml of EGF. We cultured NIH-3T3 cells (ATCC #CRL-1658) in Dulbecco's modified Eagle's medium (Invitrogen) supplemented with 10% fetal calf serum (Eurobio), 1000 U/ml penicillin/streptomycin (Invitrogen). We cultured FUCCI ESCs (kind gift from D. Landeira) as[38].

### Plasmid construction and generation of cell lines
We fused SNAP-3xHA coding sequences downstream H3.1 and H3.3 CDS[66] and inserted this fusion protein into the pBS31 vector[65] using NEBuilder® HiFi DNA Assembly Master Mix (NEB). These expression vectors (pB31-H3.1- and pB31-H3.3- SNAP-3xHA) enabled the production of H3.1- and H3.3- proteins with the tag SNAP-3xHA at their C-terminus. To generate constructs with point mutations H3.2-, H3.1-A31S- and H3.3-S31A-, H3.3-S31D- SNAP-3xHA plasmids we used site-directed mutagenesis (GenScript) with pB31-H3.1- and pB31-H3.3- SNAP-3xHA vectors, respectively. We integrated the H3-SNAP-3xHA coding sequence in KH2 mouse ESCs downstream of the Type I Collagen (Col1A1) locus containing a Frt site under the control of a TET-ON regulatory region. We co-transfected pB31-H3-SNAP-3xHA plasmids with the pCAGGS-FlpE Vector (Addgene #20733) using Nucleofector Kit 2 (Amaxa) according to manufacturer's instructions. To generate TALE-HIRA-WT-Clover and TALE-HIRA-W799A-D800A-Clover, we inserted into the pTALYM3B15 plasmid (obtained from Addgene #47878) the HIRA-WT and HIRA-W799A-D800A coding sequences[55]. using NEBuilder® HiFi DNA Assembly Master Mix (NEB) and verified all constructs by Sanger sequencing. We verified protein expression by Western blot and immunofluorescence analysis. To obtain clones that stably integrated the H3-SNAP-3xHA tag, we selected colonies on hygromycin for 14 days and isolated single

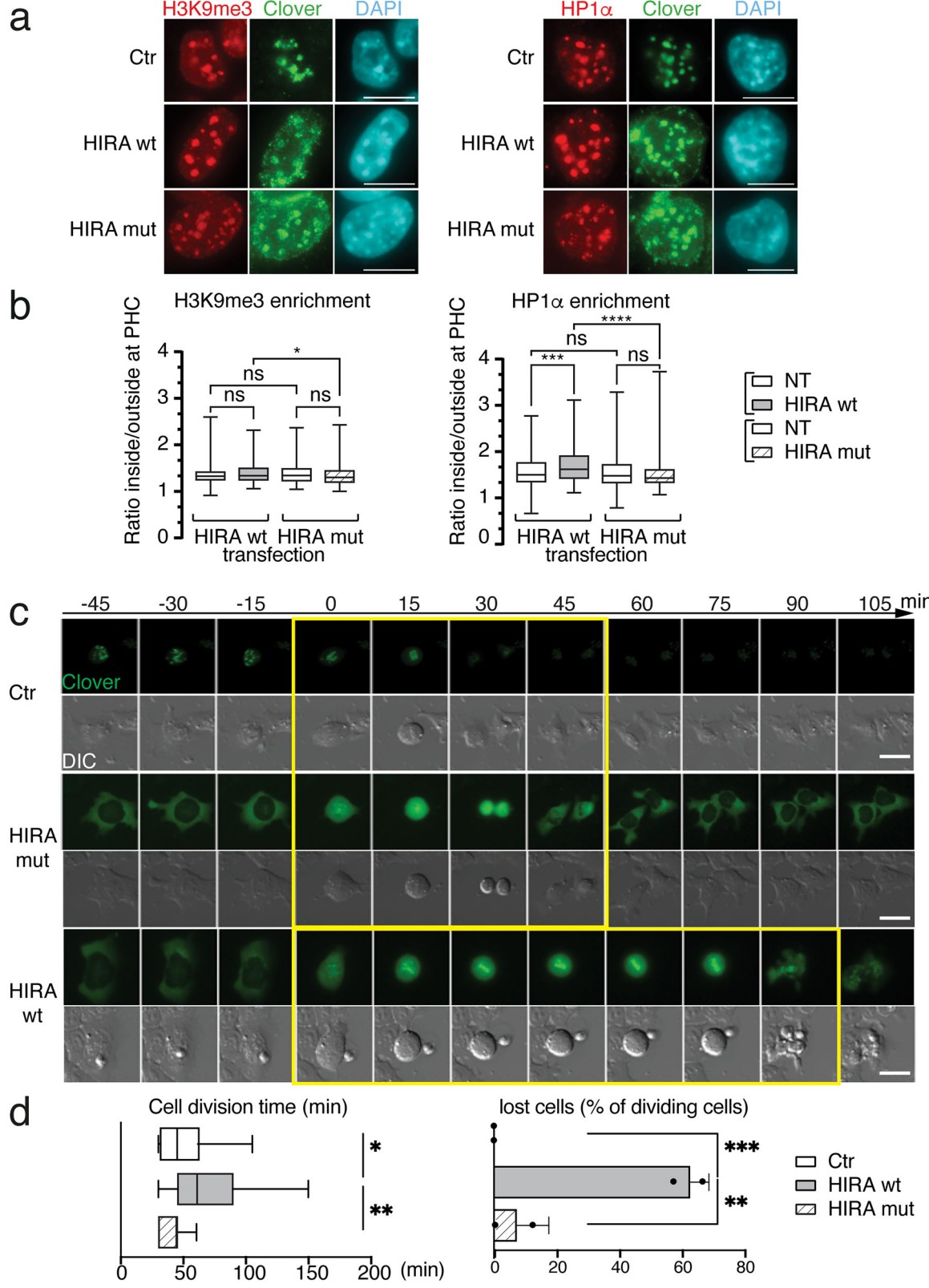

clones screened by genotyping and sequencing. We induced histone expression by adding 1 µg/ml doxycycline at least 48 h before analysis. We generated NIH-3T3 cells stably expressing either H3.1- or H3.3- SNAP-3xHA as in ref. 66 with selection using 10 µg/mL Blasticidin after retroviral transfection. We transfected ESCs with the TALE constructs using Nucleofector Kit 2 (Amaxa) according to manufacturer's instructions.

**Total cell extract preparation and western blotting**
We prepared total protein extracts by boiling the cell pellet in SDS PAGE loading buffer (Invitrogen) complemented with NuPage reducing agent (Invitrogen) and Universal Nuclease (Pierce) and performed Western blotting as in ref. 41 then acquired immunoblot images with ChemiDoc Imager (Biorad). We used the following antibodies at 1/1000 dilution: HA, Roche, #1867423; Oct3/4 # BD Biosciences 611203;

**Fig. 5 | H3.3 deposition at mouse chromocenters impacts cell viability and cell division time. a** Representative immunofluorescent images H3K9me3 (left, red) and HP1α (right, red) of ESCs expressing Ctr, HIRA wt or HIRA mut constructs (Clover, green) along with DNA staining (DAPI, cyan). Scale bars, 10 µm. **b** Boxplots of enrichment values of H3K9me3 (left) and HP1a (right) in ESCs expressing HIRA wt, HIRA mut, or non-transfected cells (NT) from 4 experiments. The center of the boxplot is the median, the bounds of the box are the 1st and 3rd quartiles and the whiskers extend to max and min. For H3K9me3 HIRA wt transfection n = 335 and 220 for NT and HIRA wt; for H3K9me3 HIRA mut transfection, n = 222 for NT and 164 for Hira mut; for HP1α HIRA wt transfection, n = 276 for NT and 178 for HIRA wt; for HP1α HIRA mut transfection, n = 385 for NT and 160 for HIRA mut. ANOVA two-way was used for statistical analysis: ns (p > 0.05), *(p < 0.05), ***(p < 0.001), **** (p < 0.0001). P values for H3K9me3 enrichment: NT(wt) vs HIRA wt = 0.1035; NT(wt) vs NT(mut) = 0.5365; HIRA wt vs HIRA mut = 0.0226; NT(mut) vs HIRA mut = 0.1521. P values for HP1α enrichment NT(wt) vs HIRA wt = 0.0004; NT(wt) vs NT(mut) = 0.1033; HIRA wt vs HIRA mut = 0.0000002676; NT(mut) vs HIRA

mut = 0.2767. Source data are provided as a Source Data file. **c** Series of 15 min time lapse images from ESCs expressing Ctr, HIRA mut and HIRA wt TALE fusions. Clover (green) and Differential Interference Contrast (DIC) images are shown. Time (top arrow) is set to 0 when cell rounds up at beginning of mitosis. Yellow boxed images correspond to cell division. Scale bars, 20 µm. **d** Quantitative analysis. Left: Boxplot of cell division time. The centre of the boxplot is the median, the bounds of the box are the 1st and 3rd quartiles and the whiskers extend to max and min. Dots indicate mean of each experiment. Data are from 31 nuclei for Ctr, 48 for HIRA wt, and 43 for HIRA mut cells per condition from 2 experiments. Mann–Whiney two-tailed test: *(p < 0.05), **(p < 0.01), ***(p < 0.001). P values Ctr vs HIRA wt = 0.0178; HIRA wt vs HIRA mut = 0.0039. Right: Proportion (in %) of "lost" after cell division. The bars represent the mean and error bars the s.d. Dots indicate mean of each experiment. Data are from 8 nuclei for Ctr, 24 for HIRA wt, and 10 for HIRA mut cells from 2 experiments. Mann–Whiney two-tailed test: *(p < 0.05), **(p < 0.01), ***(p < 0.001). P values Ctr vs HIRA wt = 0.0051; HIRA wt vs HIRA mut = 0.0224. Source data are provided as a Source Data file.

H4, Abcam #ab31830; H3.1/2, Active motif #61629; H3.3 Active motif #91191.

## Microscopy staining and acquisition

Cells seeded and grown on fibronectin-coated glass coverslips were transferred into a four-well plate (ThermoFisher Scientific) for labeling. We performed the SNAP-tag labeling in vivo with 2 µM SNAP-Cell TMR-Star (New England Biolabs). We selectively visualized histones incorporated into chromatin by extracting soluble histones with 0.5% Triton prior to fixation with 2% paraformaldehyde for 20 min as[31]. We revealed DNA synthesis by EdU incorporation and Click reaction (Click-iT EdU imaging kit, Invitrogen) as[31]. We performed Major satellite RNA FISH staining as in ref. 17. To enhance Clover detection we immunostained Clover with anti-GFP. We achieved co-staining of H3.1/2 and H3.3 by using isotype-specific secondary antibodies. We performed immunofluorescence stainings and EdU detection as in ref. 31 with the following antibodies: H3.1/2, Active motif #61629 (IgG2b); H3.3 Active motif #91191 (IgG2a); HA, Roche #1867423; Aurora B, BD transduction laboratories #611082; HP1α, Sigma #H2164; Oct 3/4 BD Biosciences #611203; H4, Abcam #ab31830; GFP in house made anti eGFP. We used AF conjugated secondary antibodies from Invitrogen.

We acquired epifluorescence images with Zeiss Imager Z1 epifluorescence microscope with 40×/1.3 NA 63×/1.4 NA 100×/1.49 NA objectives piloted with Metamorph software and an ORCA-Flash4.0 LT PLUS Digital CMOS camera (Hamamatsu). Confocal images were acquired with a confocal microscope LSM780 (Zeiss−Germany) with a 63×/1.46NA objective and an AXR confocal microscope (Nikon−Japan) with a 60×/1.42NA objective and 0.172 µm z-step. The AXR images were acquired using the NSPARC detector allowing a higher resolution. For STORM imaging, H3.1/2 and H3.3 were stained by immunofluorescence using AF647 secondary antibodies and mounted for STORM imaging as in ref. 31. STORM images were acquired with a SAFe 360 module (Abbelight, Cachan, France) in 2D single color mode using a 640 nm laser at 100% laser power (Oxxius, 540 mW) with HiLo illumination and a 100×/1.49 NA TIRF objective and a Hamamatsu Fusion BT sCMOS camera. We used Abbelight Neo software with standard parameters. 20,000 frames were acquired for each image and filtered by localization precision (25 nm), minimum blinking neighboring distance (500 nm), and merging. We performed time-lapse microscopy using the Thunder Imaging System (Leica) equipped with a Hit Stage Top Incubation System (Tokai). Briefly, we transfected cells with TALE-constructs and plated on fibronectin-coated dishes (Ibidi) and acquired images every 15 min with a 40×/0.95 NA objective, during a period of 24 h, starting 24 h after transfections.

## Microscopy images, visualization, and analysis

We used Fiji software for microscopy image visualization and time-lapse analysis. For STORM microscopy, the drift correction was done

using the Neo in-built RCC cross-correlation method with 1500 frames stack size and 25 nm lateral pixel size. A final DME algorithm[67] with an in-house adapted version in Matlab2024b using 50 frames as parameter was applied. STORM image rendering was done using 10 nm pixel size. Due to chromatic abberations, the DAPI focal plane is slightly different than the far red one (H3.1 and H3.3).

To quantify H3.1, H3.3, H3K9me3, and HP1α enrichment at chromocenters, we used the custom 3D-FIED Fiji macro[32]. We used an enrichment value of 1.2 of the Clover signal to discriminate the cells expressing the TALE fusions (positive, ratio >1.2) from the cells non-transfected (NT) non-expressing the TALE fusions (negative, ratio <1.2). For quantification of the proportion of cells with various patterns of H3 variants at chromocenters, we defined cells with pattern (i) "Enriched" when they displayed at least three chromocenters for which H3 signal intensity is higher than that of the nucleus; (ii) "Even" distribution when the H3 signal at the chromocenters was that of the nucleus; (iii) "Excluded" when the H3 signal at the chromocenters was lower than that of the nucleus. For the quantification of cells within the cell cycle, we identified S phase cells by EdU detection, G2 as EdU negative and AuroraB positive, and G1 as both EdU and AuroraB negative. For quantification of FUCCI ESCs within the cell cycle, we defined Early G1 phase cells as positive for hCdt1 detection, late G1/early S positive both hCdt1 and Geminin, mid/late S Geminin positive, and G2 cells with the strongest Geminin signal. All plots and data visualization are generated with GraphPad and Python.

## H3.1- and H3.3- SNAP capture-seq

We performed H3 SNAP ChIP-Seq of native nucleosomes (SNAP capture-seq) in ESCs and NPC cells by using the SNAP-capture procedure as in refs. 33,34. We induced synthesis of H3-SNAP-Tag in ESCs and NPCs by adding 1 µg/ml doxycycline before cell collection. We used 4 million of cells processed 1 million at a time as in ref.[34]. Briefly, nuclei were digested with micrococcal nuclease and soluble native nucleosomes were collected as supernatant following centrifugation 10,000 × g 10 min (input). SNAP-tagged nucleosomes were purified by overnight incubation with SNAP capture beads (S9145S, NEB) (SNAP-captured). DNA was extracted from nucleosomes (Input and SNAP-captured) by ProteinaseK/SDS treatment as in ref. 34. We prepared sequencing libraries at the Next Generation Sequencing (NGS) platform at Institut Curie (Illumina TruSeq ChIP kit) and performed PE100 sequencing on Illumina NovaSeq 6000. Sequencing data was processed from raw fastq files as described in ref. 33 (Seq data deposited in ArrayExpress under accession code E-MTAB-14240. Briefly, we mapped reads to the soft-masked mouse reference genome (GRCm38, release 98) from Ensembl with bowtie2 v2.3.4.2[68] very-sensitive option. We sorted, flagged duplicates, and indexed BAM files with SAMtools v1.9[69]. We used repetitive element annotation from Ensembl (mus musculus core 102_38) and filtered out tandem ("trf")

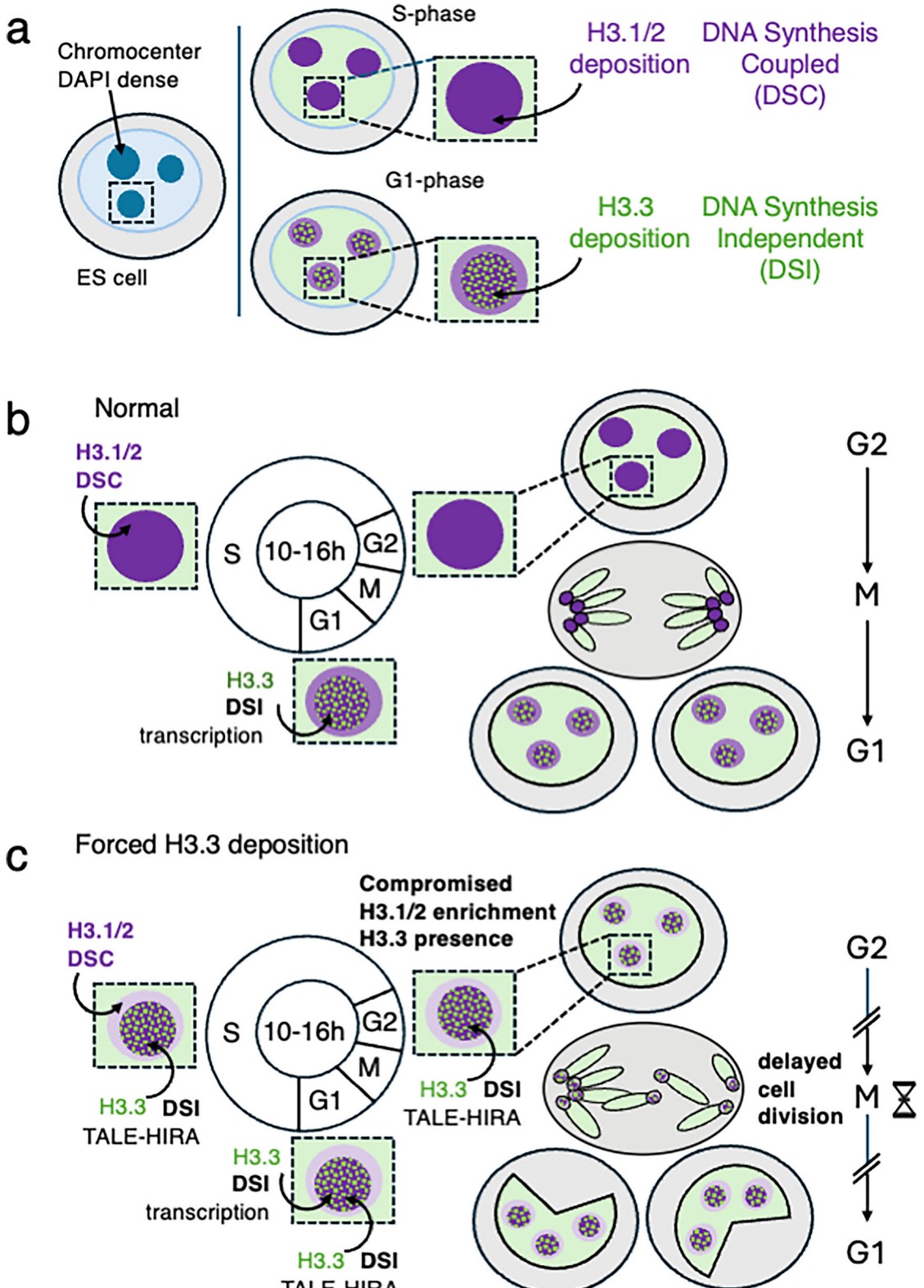

**Fig. 6 | Schematic model for H3 variant dynamics at chromocenters. a** H3.1/2 and H3.3 variants deposition at chromocenters depends on cell cycle and distinct deposition mechanisms, DNA Synthesis Coupled (DSC) and DNA Synthesis Independent (DSI), respectively. **b** In a normal ES cell cycle, during G1 phase as the increased transcriptional activity in chromocenter promotes H3.3 deposition by DSI allowing to counteract H3.1/2 enrichment. In S phase, during chromocenter

replication, H3.1/2 enrichment is re-established and maintained in G2 prior to cell division. **c** Forced deposition of H3.3 by DSI throughout cell cycle by targeting HIRA to chromocenter (TALE-HIRA) results in continuous H3.3 deposition and compromises H3.1/2 enrichment at chromocenter. This interference in turn leads to defects in nuclear morphology and cell cycle division.

and low complexity ("dust") repeats classes. For each sample, the number of reads at each repeat was calculated by overlapping the repeat annotation with the genomic coordinates of fragments mapped in pairs, excluding duplicates, extracted from the BAM files. Samples were normalized to the total number of reads mapped to all repeats (CPM), then by dividing by repeat length, and finally by dividing by the matching input sample. Then, the IP to input ratio was $\log_2$-transformed, and to allow comparison between conditions, the cross-sample was normalized by computing z-scores. We quantified the number of normalized reads mapping to the GSAT_MM (Major Satellite) annotations (a total of 72) from each sample.

### Reporting summary

Further information on research design is available in the Nature Portfolio Reporting Summary linked to this article.

## Data availability

The SNAP ChIP-seq data generated in this study have been deposited in ArrayExpress under accession code E-MTAB-14240 https://www.ebi.ac.uk/biostudies/ArrayExpress/studies/E-MTAB-14240. Source data are provided with this paper.

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

## Acknowledgements

We thank E. Boyarchuk for starting the project, B. Kaffe for generating PBS31_H3.1-SNAP-3xHA plasmid, and D. Landeira for the FUCCI ESCs. We thank D. Ray-Gallet and D. Jeffery for critical reading, and all team members and the unit for discussions. We acknowledge the Cell and Tissue Imaging Platform PICT-IBiSA (member of France-Bioimaging ANR-10-INBS-04) PICT@Pasteur and PICT@BDD, Nikon Imaging Centre @ Institut Curie-CNRS (part of PICT-IBiSA), and ICGex NGS platform of the Institut Curie. Funding to G.A. includes La Ligue Nationale contre le Cancer (labellisation), Labex DEEP-PSL (ANR-11LABX-0044_DEEP, ANR-10-IDEX-0001-02), ERC-2015-ADG694694 ChromADICT, and Horizon EIC Pathfinder project 101099654 "RT-SuperES". Funding to E.M. includes Israel Ministry of Science (MOST-DKFZ) collaborative program ([0005358 to E.M.) and Horizon EIC Pathfinder project 101099654 "RT-SuperES". S.A. benefited from H2020 MSCA-ITN—EpiSyStem (Grant No. 765966), Fondation Recherche Medicale (FRM) (Grant No. FDT202106012804), and EIC Pathfinder project 101099654 "RT-SuperES". T.K. benefited from H2020 MSCA-ITN—ChromDesign. D.B. benefited from EIC Pathfinder project 101099654 "RT-SuperES".

## Author contributions

G.A., J.P.Q., and S.A. conceived the strategy and wrote the manuscript. G.A. and JP.Q. supervised the work. S.A. performed experiments and analyzed data with J.P.Q. A.F. and H.M. experiment for Fig. 1f. T.K. analyzed data in Fig. 1f. D.B. for Fig. S4 and P.L.B. for Fig. S2a. D.M. and M.G. analyzed STORM data for Fig. 1d. E.M. advised on stem cell biology. G.A. acquired funding. Critical reading and data discussion involved all authors.

## Competing interests

The authors declare no competing interests.
