## [Transparent Peer Review file · Nature Communications]

H3.3 deposition counteracts the replication-dependent enrichment of H3.1 at chromocenters in embryonic stem cells

Corresponding Author: Dr Geneviève Almouzni

Version 0:

Reviewer comments:

Reviewer #1

(Remarks to the Author)

In this paper, Arfe et al. demonstrate that canonical histone H3.1 is preferentially incorporated into chromocenters relative to histone variant H3.3. While the observations are interesting, the broader implications, conclusions, or mechanisms are not fully explored, making it challenging to understand the full significance of the findings. Please find attached specific points that need to be addressed.

1) The nuclear distribution of H3.2 and H3.3 variants in different cellular states is examined, but the study could benefit from a deeper exploration of the broader implications or mechanisms behind these observations.

It has been reported that H3.1 variants are enriched at chromocenters (although not thoroughly investigated), while H3.3 is typically excluded and found mostly in active chromatin. The study reaffirms this but could further elaborate on the underlying reasons or broader biological significance. The default assumption is that H3.1/2 are preferentially incorporated due to their overwhelming presence compared to H3.3, except under specific circumstances like chromocenter formation in early mouse embryos where H3.3 accumulates. However, the study does not address how H3.3 counteracts H3.1 enrichment at chromocenters. The role of the reciprocal behaviour between H3.1 and H3.3 during cell differentiation remains unexplored, leaving a gap in understanding the dynamics of histone variant distribution and its implications for chromatin structure and function.

2) Figure 1. The sequencing data (Fig. 1e) indicated that H3.1 is preferentially deposited at chromocenters relative to H3.3. However, the data could not be located at the accession supplied (E-MTAB-10619) which only contained data for cervical adenocarcinoma cell lines, making it challenging to locate the specific data discussed. In addition, the methods provided did not adequately describe the experiment and are not sufficient to reproduce the experiment.

3) Figure 2. During cell cycling, H3.1 enriches at pHC from mid S to G2 (less enriched at G1/early S), whereas H3.3 shows the most exclusion at these cell cycle stages. Based on these findings., it is proposed that H3.1 enrichment depends on S phase DNA synthesis coupled (DSC) deposition and competition against H3.3 deposition. These findings and hypotheses have not been fully explored. For examples,

What drives the different patterns of H3.1 and H3.3 enrichment (or DNA synthesis independent or DSI H3.3 deposition) at pHC during the G1/early S phase in ESCs and NPC/3T3 cells? Transcription has been reported at gamma satellite in late G1 phase, with their synthesis decreasing during mid S phase, which coincides with pHC replication. Could pHC transcription affect the dynamics of H3.3/H3.1 enrichment at pHC? Will the depletion of Suv39h1/2 methyltransferases (responsible for H3K9me3) affect H3.3/H3.1 enrichment at pHC? Perhaps the level of pHC transcription may also differ between ESCs, NPCs, and 3T3 cells.

Does the depletion of H3.3 deposition lead to increased H3.1/2 enrichment at pHC, or does the depletion of H3.1/2 results in increased H3.3 deposition? Does the over expression of H3.3 and its chaperones (without targeting using the TALEN system) force the exclusion of H3.1/2 deposition?

4) Figure 4. The rationale for using a HIRA-TALEN fusion to incorporate H3.3 at pericentric chromocenters is not entirely clear, especially since H3.3 is typically incorporated by the DAXX-ATR complex, as mentioned in the introduction: "In contrast, H3.3, a replacement variant, is incorporated in a DNA Synthesis-Independent (DSI) fashion either by the

chaperone HIRA at active chromatin regions and specialized nuclear domains or by the DAXX-ATR complex at constitutive heterochromatin regions, including telomeres, retrotransposons, and pericentric heterochromatin." Given that HIRA and DAXX-ATR likely associate with different modifying complexes, it is possible that forcing HIRA, rather than DAXX-ATR, into "silenced" pericentric chromocenters has an effect. Does the targeting of HIRA affect pHC transcription and chromatin status, which have not been addressed? Will targeting of DAXX-ATR (instead of HIRA) have the same effect on H3.3/H3.1 enrichment? The specific pathways and mechanisms potentially disrupted by the incorporation of HIRA-mediated H3.3 at chromocenters remain unclear and have not been addressed in this study.

Minor comments:

Could the color selection for "Even" and "Excluded" be made more distinct? Currently, both are grey (and are quite similar), making it difficult to differentiate them.

Reviewer #2

(Remarks to the Author)

This manuscript by Arfè et al shows that H3.1/3.2 are preferentially enriched at pericentric heterochromatin in mouse cells compared to H3.3. The authors pinpoint the relevant localization motif to the SVM stretch of H3.1/3.2. By targeting the H3.3 chaperone HIRA to PCH via TALEN coupling, the authors show that H3.1 deposition along with cell division are compromised. The study is designed very well and the conclusions are in general supported by the data. The main weaknesses are the low resolution of microscopy images and the absence of other types of data supporting image-based conclusions (except Fig 1e).

Main comments

- If I understood correctly, the images presented in the paper were not acquired via confocal microscopy. Although the epifluorescence images mostly did serve the purpose, they also limited the strength of the drawn conclusions. I appreciate that it would be difficult to repeat everything, however for some aspects I would encourage the authors to add higher resolution and/or higher precision images, eg. For Figure 4.
- In Figure 3, the authors perform a nice experiment to delineate the sequence specificity of H3.1/3.2/3.3. However no data are provided as to the expression levels or integration levels of these histones to PCH. Since the readout is global PCH localization, one would need to make sure that the expressed histones are actually integrated at some level at these sites. Otherwise the conclusions may be invalid.
- In Figure 4, the authors show successful localization to PCH of all TALE constructs. However, as the authors also point out, H3.3 deposition cannot be deduced from the microscopy images. The one cell that is presented shows intracellular variability. The authors should provide ChIP data to show that H3.3 deposition is working. qPCR with maj sat primers would fully answer this point.
- In Figure 5, please also show images of the control TALE constructs.
- The authors claim chromosome segregation defects in Figure 5 but actually do not show this. I can see from the images that cell division time is longer in the HIRA OE cells. However I cannot find any data related to cell division defects. The authors should provide higher resolution chromosome segregation data to support this claim.
- Bringing Figure 5a/b and c together, I do not fully understand why the chromosomes would fail to segregate. HP1 as well as H3K9me3 (by tendency) are enriched more at PCH in the HIRA construct, which in theory should establish even more properly controlled PCH domains. H3.3 has higher turnover rate at chromatin, is this the problem?
- In Figure 5e, how was cell death measured? Chromosome segregation defects do not lead to cell death immediately. How long after the division is cell death measured?

Minor comments

- Are the data in Figure 1e significant? The 'n' number is not clear to me. The legend states 'n represents the number of repeat elements at each differentiation step'. Since the authors look at maj sat repeats, for which I assume consensus sequence primers are used, what does the n mean here?
- The terminology in the labeling of Figure 4d is confusing. The W799A-D800A+ reads as W799A negative D800A positive.
- In all quantification figures, please state the number of cells used for quantification and not only %.
- There are several typos in the text and legends, therefore the authors should once again carefully go through the text.

Reviewer #3

(Remarks to the Author)

I co-reviewed this manuscript with one of the reviewers who provided the listed reports. This is part of the Nature Communications initiative to facilitate training in peer review and to provide appropriate recognition for Early Career Researchers who co-review manuscripts

Version 1:

Reviewer comments:

Reviewer #1

(Remarks to the Author)

The authors have addressed my primary concerns. However, I have a few minor comments that should be addressed to further strengthen the manuscript.

While I now understand the rationale behind selecting HIRA to target H3.3 to the chromocenter, I believe it is important for the authors to explain this reasoning more explicitly in the text. Additionally, the authors may consider discussing whether artificial targeting using the ATRX/DAXX complex (which could recruit H3K9me3/HP1 α) might result in a different outcome, particularly in terms of promoting the reduction of H3.1 deposition and inducing changes in nuclear morphology. A discussion on this connection would enhance the overall conclusions.

It is interesting that wild-type HIRA targeting did not significantly affect major satellite transcription. Does this truly suggest that transcription can be bypassed and that HIRA-mediated H3.3 deposition alone is sufficient to challenge H3.1 deposition at the chromocenter? The authors may consider discussing their findings in the context of observations made in SUVAR3-9-depleted cells, where increased major satellite transcription and enhanced H3.3 deposition have been observed, accompanied by a reduction in H3.1 deposition. This comparison could provide valuable insights into the underlying mechanisms of histone deposition dynamics. Additionally, the enrichment of labile H3.1 and the turnover of H3.1/H3.3 at the chromocenter in ESCs may still be transcription-dependent, a point worth exploring further.

I agree with the authors that the choice of histone chaperones involved in H3.1/H3.3 deposition may be a key determinant of nucleosomal composition. Expanding on this point could provide additional depth to the discussion.

Finally, the label on Figure S5d appears to be incorrect and should be corrected.

Reviewer #2

(Remarks to the Author)

The authors addressed my comments by adding higher precision data and toning down the comments that were not sufficiently supported by the data. I do not have other concerns.

Point by point answer to reviewer comments

Reviewer #1 (Remarks to the Author):

In this paper, Arfe et al. demonstrate that canonical histone H3.1 is preferentially incorporated into chromocenters relative to histone variant H3.3. While the observations are interesting, the broader implications, conclusions, or mechanisms are not fully explored, making it challenging to understand the full significance of the findings. Please find attached specific points that need to be addressed.

We thank this reviewer for acknowledging the interest of our observations and for giving us opportunities to explore further their broader implications and underlying mechanisms. In the revised version, we now provide novel data based on reviewer's comments. Importantly, we explored in more depth the mechanisms leading to distinct H3.1 and H3.3 accumulation at chromocenters (see our comments and new data below). We now discuss these points in the main text, and we changed our model accordingly to take these aspects into consideration as shown in our revised figure 6. This has certainly improved our manuscript and increased its potential impact in the context of cell cycle, cell fate and development.

1) The nuclear distribution of H3.2 and H3.3 variants in different cellular states is examined, but the study could benefit from a deeper exploration of the broader implications or mechanisms behind these observations.

It has been reported that H3.1 variants are enriched at chromocenters (although not thoroughly investigated), while H3.3 is typically excluded and found mostly in active chromatin. The study reaffirms this but could further elaborate on the underlying reasons or broader biological significance. The default assumption is that H3.1/2 are preferentially incorporated due to their overwhelming presence compared to H3.3, except under specific circumstances like chromocenter formation in early mouse embryos where H3.3 accumulates. However, the study does not address how H3.3 counteracts H3.1 enrichment at chromocenters. The role of the reciprocal behaviour between H3.1 and H3.3 during cell differentiation remains unexplored, leaving a gap in understanding the dynamics of histone variant distribution and its implications for chromatin structure and function.

Based on this reviewer's comments and the second reviewer's useful suggestions, we have further explored how H3.1 and H3.3 could accumulate at chromocenters. Our analysis now includes higher resolution with both confocal imaging (new fig. S2a) and STORM super resolution (new fig. 1d) to characterize the distinct patterns for H3.1 or H3.3 under various conditions. These data are presented and discussed in the revised manuscript. They also led us to modify our final figure to propose a model (revised figure 6) in which we stress the dominant effect of the H3.3 deposition,

transcription of major satellite repeats and the embryonic cell cycle dimension which all prove important when considering cell state and differentiation.

Indeed, major satellite repeats RNA levels are substantially higher in ESCs relative to most other cell types (Efroni et al., 2008) (Martens et al., 2005) (Novo et al., 2016) (Percharde et al. 2017) (Tosolini et al., 2018). Interestingly, our findings show that H3.1 enrichment accumulation is consolidated in NPC and 3T3 cells when compared to ES cells (fig. 1e, fig. 2c, fig S2c). This consolidation is also confirmed in our ChIP experiments where we find that more H3.1 associate with major satellite in NPC compared to ES cells (figure 1f). Mechanistically, the key question was whether it is the transcription that directly allows to evict H3.1, and/or whether it was possible to directly bring H3.3 to dominantly remove H3.1. To address this issue, we refined our work and repeated a key experiment, where we tested whether an artificial placement of H3.3 could promote H3.3 deposition and counteract H3.1 accumulation (new fig. S4). All the data are consistent.

The results support a model in which the aberrant provision of H3.3 (even modest) interferes severely with H3.1 accumulation. This finding is important for a general view concerning chromatin dynamics as follow. H3.1 accumulation at chromocenters represents a default state. Its deposition linked to S phase occurs via the replication dependent mechanism. The aberrant provision of H3.3 outside its normal time window functions in a dominant manner and allows to kick out H3.1 deposited in S phase. Indeed, H3.1 enrichment at chromocenters is decreasing (fig 4b right, 4c) when chromocenters are targeted by HIRA H3.3 deposition (new figure S4). Conversely, H3.3 is less depleted (figure 4b left). It is interesting though to note that compromising the CAF-1 dependent deposition pathway can also provide opportunities for H3.3 deposition to kick in (Ray-Gallet et al., 2011).

2) Figure 1. The sequencing data (Fig. 1e) indicated that H3.1 is preferentially deposited at chromocenters relative to H3.3. However, the data could not be located at the accession supplied (E-MTAB-10619) which only contained data for cervical adenocarcinoma cell lines, making it challenging to locate the specific data discussed. In addition, the methods provided did not adequately describe the experiment and are not sufficient to reproduce the experiment.

We apologise for the mistake; we now provide the correct accession number: E-MTAB-14240. Below is an access link for the reviewers (data will be made available broadly upon publication of the manuscript).

<https://www.ebi.ac.uk/biostudies/arrayexpress/studies/E-MTAB-14240?key=6b462a44-b73e-42d7-8eb9-8e491a8786f0>

We have now also described the methods in detail along with the deposited data and a detailed protocol is available in Forest et al., 2024 to which we now clearly refer to.

3) Figure 2. During cell cycling, H3.1 enriches at pHC from mid S to G2 (less

enriched at G1/early S), whereas H3.3 shows the most exclusion at these cell cycle stages. Based on these findings., it is proposed that H3.1 enrichment depends on S phase DNA synthesis coupled (DSC) deposition and competition against H3.3 deposition. These findings and hypotheses have not been fully explored. For examples,

What drives the different patterns of H3.1 and H3.3 enrichment (or DNA synthesis independent or DSI H3.3 deposition) at pHC during the G1/early S phase in ESCs and NPC/3T3 cells? Transcription has been reported at gamma satellite in late G1 phase, @with their synthesis decreasing during mid S phase, which coincides with pHC replication. Could pHC transcription affect the dynamics of H3.3/H3.1 enrichment at pHC? Will the depletion of Suv39h1/2 methyltransferases (responsible for H3K9me3) affect H3.3/H3.1 enrichment at pHC? Perhaps the level of pHC transcription may also differ between ESCs, NPCs, and 3T3 cells.

Does the depletion of H3.3 deposition lead to increased H3.1/2 enrichment at pHC, or does the depletion of H3.1/2 results in increased H3.3 deposition? Does the over expression of H3.3 and its chaperones (without targeting using the TALEN system) force the exclusion of H3.1/2 deposition?

We thank the reviewer for these comments which prompted us to investigate in more depth which mechanism promotes H3.3 deposition at chromocenters. We followed the useful suggestion, and in the revised version we now used Suv39h1/2 double null ES cells (Suv39dn ES cells) to explore how H3.1 and H3.3 behave at chromocenters. We found that H3.1 enrichment decreased in Suv39dn ES cells compared to wild type ES cells. This new data is now provided as a new paragraph in the results section and new figure 3d. Given that major satellite transcription at chromocenter is higher in Suv39dn versus wt ES cells (Camacho et al., 2017) (Martens et al., 2005), our findings suggest a potential link between H3.3 deposition and transcription as proposed by the reviewer. Thus, transcription at chromocenter correlates with a decrease in H3.1 enrichment, in line with some H3.3 deposition at those sites. Interestingly, this correlation is further reinforced considering that during G1/early S phase, when major (gamma) satellite repeats transcription increases (Lu and Gilbert, 2007), we also found a decrease in H3.1 enrichment at chromocenters (Fig. 2c).

Furthermore, given the fact that for the H3 variants to accumulate at chromocenter in ES cells an intact SVM motif (typical of the DNA synthesis coupled deposition pathway) proved necessary while the AIG (typical of DNA synthesis independent deposition pathway) or onco-histone mutations were dispensable (Fig. 3bc, Fig. S3), we could refine the relative importance of the deposition pathways.

Based on the new data with the Suv39 dn ES cells and the above findings, we now propose a model in the revised figure 6 in which H3.1 and H3.3 levels at chromocenters depend on the cell cycle in a timely manner linked to two assembly pathways: for H3.1 deposition, the use of the DNA synthesis Coupled (DSC) pathway during mid S-phase when chromocenter replicates and for H3.3 deposition, the use of the DNA synthesis independent (DSI) that can be coupled to transcription normally

operating in G1. The combination of cell cycle, transcription and mode of deposition then lead to distinct accumulation at chromocenters.

Therefore, while H3.1 enrichment depends on the DSC deposition pathway in S phase. Yet, transcription of major satellites can provide possibilities for H3.3 deposition via (i) increased frequency/opportunity of naked DNA exposure, or (ii) through the link with Pol-II (Ray-Gallet et al., 2011). Notably, a HIRA dependent pathway could exploit either gap-filling or transcription coupled deposition. In this way the passage of the polymerase would evict pre-existing histones and put in place H3.3 (see (Torné; et al., 2020)). This could lead to a decrease of H3.1 at the time of a transcription burst. We now discuss this important aspect in the revised version in line with differences in Maj Sat transcription for the various differentiation states (Efroni et al., 2008) (Martens et al., 2005) (Tosolini et al., 2018). For the transcription of MajSat at pCH during the cell cycle, indeed the levels of transcription have been reported (Lu and Gilbert, 2007). This could indeed explain the change in variant enrichment that we detected in the different cell types that we analyzed.

4) Figure 4. The rationale for using a HIRA-TALEN fusion to incorporate H3.3 at pericentric chromocenters is not entirely clear, especially since H3.3 is typically incorporated by the DAXX-ATRX complex, as mentioned in the introduction:

“In contrast, H3.3, a replacement variant, is incorporated in a DNA Synthesis-Independent (DSI) fashion either by the chaperone HIRA at active chromatin regions and specialized nuclear domains or by the DAXX-ATRX complex at constitutive heterochromatin regions, including telomeres, retrotransposons, and pericentric heterochromatin.”

Given that HIRA and DAXX-ATRX likely associate with different modifying complexes, it is possible that forcing HIRA, rather than DAXX-ATRX, into "silenced" pericentric chromocenters has an effect. Does the targeting of HIRA affect pHC transcription and chromatin status, which have not been addressed? Will targeting of DAXX-ATRX (instead of HIRA) has the same effect on H3.3/H3.1 enrichment? The specific pathways and mechanisms potentially disrupted by the incorporation of HIRA-mediated H3.3 at chromocenters remain unclear and have not been addressed in this study.

We should have explained better our rationale. Indeed, we wished to assess how a “forced” deposition of H3.3 by a chaperone that do not “normally act at the site” in a WT background, rather than overexpressing a histone chaperone already present at pHC such as DAXX/ATRX. The TALE-HIRA fusion that we used fulfilled these requirements, to challenge a situation and perturb chromatin dynamics (in this case, H3.1/H3.3 enrichment). These perturbations were key to further develop our imaging analysis pipeline.

This approach proved successful since it allowed us to alter H3.1/H3.3 enrichment at pCH. In the revised version of this manuscript, we repeated these experiments and

carried out further analysis. We now provide new images for H3.1 and H3.3 detection at chromocenter acquired with confocal microscopy (new fig. S4). These data show that when HIRA is targeted to chromocenters the depletion of H3.3 normally observed is challenged and less pronounced (Fig. 4a arrows, Fig. 4b left, new figure S4). Of course, given the high levels of H3.3 outside chromocenters, measuring a true enrichment is tedious (Fig 4c left). See also response to reviewer#2 below.

We also investigated whether targeting HIRA-mediated H3.3 deposition at chromocenter could affect transcription of major satellite DNA repeats (MajSat). First, we used RT-qPCR to measure MajSat transcription and did not detect increased amounts of MajSat in cells transfected with TALE-HIRA wt clover (HIRA wt) compared to the controls TALE-clover (Ctr) and TALE HIRA mut clover (HIRA mut) (figure for reviewer only below this comment). However, this could be due to the fact that all cells are not transfected and an analysis in bulk could dilute or mask an impact of the TALE-HIRA wt clover. Thus, we used a microscopy-based approach enabling to select transfected cells expressing TALE fusions (see new Fig. 4C top, clover positive) and monitored in those cells MajSat RNA by RNA FISH (new fig. S5a). Most importantly we did not detect a specific increase for the TALE-HIRA wt clover compared to the TALE constructs controls (TALE-clover and TALE-HIRA mut clover) (new Fig. S5b). Furthermore, within the cells positive for MajSat foci, we did not detect an increase in the number of foci per nucleus for the TALE-HIRA wt clover compared to the TALE constructs controls (TALE-clover and TALE-HIRA mut clover) (new Fig. S5c). Together these data indicate that under our experimental conditions the targeting of HIRA at pHC does not affect MajSat transcription. This is also interesting because it shows that we can bypass the transcription and that wt HIRA proficient for H3.3 deposition is sufficient to challenge H3.1 at chromocenters without the transcription.

Quantification of major RNA by RT-qPCR. Data is normalized to 1 for the TALE-clover transfected cells. Average and SD from 7 experiments.

Minor comments:

Could the color selection for "Even" and "Excluded" be made more distinct?

Currently, both are grey (and are quite similar), making it difficult to differentiate them.

We thank the reviewer for this comment. In the revised version, we changed the color / grey levels to facilitate reading of the figures 1e, 2c, 2f, 3c, and S3. "Enriched" is

now displayed in black, “Even” in grey and “Excluded” in white.

Reviewer #2 (Remarks to the Author):

This manuscript by Arfè et al shows that H3.1/3.2 are preferentially enriched at pericentric heterochromatin in mouse cells compared to H3.3. The authors pinpoint the relevant localization motif to the SVM stretch of H3.1/3.2. By targeting the H3.3 chaperone HIRA to PCH via TALEN coupling, the authors show that H3.1 deposition along with cell division are compromised. The study is designed very well and the conclusions are in general supported by the data. The main weaknesses are the low resolution of microscopy images and the absence of other types of data supporting image-based conclusions (except Fig 1e).

We thank this reviewer for this positive evaluation of our work. Thanks to his/her suggestions, in the revised manuscript, we now provide microscopy images of higher resolution including both confocal images (new Fig. S2a and S4) and Super-Resolution images by STORM (new Fig. 1e). See below the details in our response to specific comments. Furthermore, based on reviewer 1's comments we also performed additional experiments enabling to deepen our mechanistic analysis and leading to a refined model as now depicted in the new Figure 6.

Main comments

- If I understood correctly, the images presented in the paper were not acquired via confocal microscopy. Although the epifluorescence images mostly did serve the purpose, they also limited the strength of the drawn conclusions. I appreciate that it would be difficult to repeat everything, however for some aspects I would encourage the authors to add higher resolution and/or higher precision images, eg. For Figure 4.

The images presented in the manuscript were acquired with classical widefield microscopy. We noticed that when using classical wide field microscopy with illumination and detection through the sample (as in our manuscript) it proved difficult to monitor changes in the non-enriched/depleted H3.3 signal at chromocenters (even when using deconvolution). However, we could detect and quantify H3.3 signal at chromocenter from live observation at the microscope as patterns Enriched, Even and Excluded (Fig. 1e, 2c, 2f, 3c, 4b and S3). We agree with the reviewer that providing images to illustrate the quantifications of these patterns is helpful.

It was satisfying to realize that when using a microscopy system with different illumination/detection (confocal and HILO illumination with STORM Super Resolution) we could achieve a better imaging of H3.3 depletion/weak signal at chromocenters. This has allowed to document our observations and provide illustrations. These new experiments and corresponding data are now reported in new Fig. S2a related to figure 1, and new Fig. S4 related to figure 4. We thank the reviewer for this useful suggestion which has significantly strengthened our data.

- In Figure 3, the authors perform a nice experiment to delineate the sequence specificity of H3.1/3.2/3.3. However no data are provided as to the expression levels or integration levels of these histones to PCH. Since the readout is global PCH localization, one would need to make sure that the expressed histones are actually integrated at some level at these sites. Otherwise the conclusions may be invalid.

We should have stressed that for all our analyses, we systematically used extraction prior to fixation to ensure that soluble histones are discarded and the chromatin fraction enriched. This includes both visualization of endogenous histones with antibodies and the exogenous SNAP-tagged histones. Thus, under these conditions, we feel confident that we can measure histones that are stably incorporated at pCH and outside pCH. For clarity, this is emphasized in the main text (“We next monitored the incorporation of these exogenous H3.1 and H3.3 into chromatin by pre-extracting soluble histones before fixation (Clément; et al., 2018) (Martini; et al., 1998) (Fig 1a, b).”) and details are provided in the method section (“We selectively visualized histones incorporated into chromatin by extracting soluble histones with 0.5% Triton prior to fixation with 2% paraformaldehyde for 20’ as (Clément et al., 2018)”.)

- In Figure 4, the authors show successful localization to PCH of all TALE constructs. However, as the authors also point out, H3.3 deposition cannot be deduced from the microscopy images. The one cell that is presented shows intracellular variability. The authors should provide ChIP data to show that H3.3 deposition is working. qPCR with maj sat primers would fully answer this point.

The point of the reviewer is well taken and indeed, we also envisaged to use a ChIP approach but given that only a small proportion of cells are transfected and display TALE localization to pCH, the ChIP signal from bulk population would account mostly from non-transfected cells. We thus tried to sort the transfected cells from the bulk population by FACS based on the clover fluorescence (TALE-clover fusions). Unfortunately, this approach due to the low signal in fluorescence for the nuclear staining failed to properly gate from the control/non transfected cells. We thus used again a microscopy-based approach to select for TALE constructs expressing cells

(clover positive, see new top panel in Fig. 4c) and used confocal microscopy to visualize changes in H3.3 signal at pCH (see new Fig. S4). We repeated the experiment for figure 4 and acquired images with confocal microscope. We found that when HIRA wt is targeted to chromocenter H3.3 is less depleted and even becomes “relatively” enriched at chromocenter and colocalizes with H3.1 (new Fig. S4). In contrast in the controls (non-transfected cells, TALE-clover and TALE-HIRA mut clover) in which H3.3 deposition is not promoted, we do not see such changes at chromocenters. Thus, targeting HIRA at pCH can lead to an increased deposition of H3.3 locally. This new data is now provided as new Fig S4 related to figure 4.

- In Figure 5, please also show images of the control TALE constructs.

We now provide in the revised Fig. 5a images of the TALE-clover constructs.

The authors claim chromosome segregation defects in Figure 5 but actually do not show this. I can see from the images that cell division time is longer in the HIRA OE cells. However I cannot find any data related to cell division defects. The authors should provide higher resolution chromosome segregation data to support this claim.

We agree since we did not provide data in direct support of chromosome segregation defects, we thus removed/modified this sentence in the revised manuscript and also amended the model to accurately reflect our findings.

To describe and illustrate the defects more accurately, we monitored by time-lapse (Fig. 5c, 5d) and we quantified by microscopy the proportion of nuclei with altered morphology in the population of cells expressing TALE-Hira wt clover (HIRA wt) and the controls TALE-clover (Ctr) and TALE-HIRA mut clover (HIRA mut). These altered morphologies include heterogenous shapes, bigger nuclear size, multilobed in line with a defective mitosis (new figure S6a). We found that the proportion of such nuclei is increased with the TALE HIRA wt construct (new figure S6b). This novel data is presented in the revised version as new figure S6.

- Bringing Figure 5a/b and c together, I do not fully understand why the chromosomes would fail to segregate. HP1 as well as H3K9me3 (by tendency) are enriched more at PCH in the HIRA construct, which in theory should establish even more properly controlled PCH domains. H3.3 has higher turnover rate at chromatin, is this the problem?

Concerning HP1 enrichments issues, we now refer to the work from Peter Rugg-Gunn lab where by depleting MSR transcripts they could trigger a more compact heterochromatin state in ES cells, i.e. resulting in a higher HP1 and H3K9me3 concentration at pHC, similar to that of differentiated cells (Joron et al., 2023) (Novo et al., 2022). Importantly, under these conditions they also observed an increase in

chromosome instability, elevated DNA damage, and defective mitosis (Novo et al., 2022). Thus, perturbing chromocenter organization by increasing heterochromatin compaction could potentially interfere with DNA repair machinery or with the kinetochore assembly at these regions. We hope that these explanations can help.

- In Figure 5e, how was cell death measured? Chromosome segregation defects do not lead to cell death immediately. How long after the division is cell death measured?

In Figure 5c, we performed a 24-hour time-lapse microscopy and followed clover positive cells. Cell division is referred to as the time spent from prophase, when cell rounds-up (1 cell) to the end of telophase when the two daughter cells re-flatten to the plate (2 cells). The time spent during these two events is what we show in Fig. 5d.

We noticed that some cells do enter in prophase, detach from the dish, but do not divide and finally disaggregate. Figure 5e 'Cell death after cell division' referred to such events. We realize that it may be confusing and now changed the formulation in text and figure and replaced cell death with "lost cells". We amended the text accordingly

Minor comments

- Are the data in Figure 1e significant? The 'n' number is not clear to me. The legend states 'n represents the number of repeat elements at each differentiation step'. Since the authors look at maj sat repeats, for which I assume consensus sequence primers are used, what does the n mean here?

We estimated the significance of the changes using 2-tailed Mann-Whitney U test and showed the results in Figure 1e. We realize that this aspect was unclear in the figure legend and changed it in the revised version: "Quantification of H3 variant enrichment at Major Satellite repeat elements in ESCs and NPCs by ChIP-Seq (SNAP-capture). The H3 variant enrichment is displayed as a Z-score of log2 enrichment of IP over input indicating enrichment when above 0 as indicated by red dotted line." Further clarification has been added in the methods.

- The terminology in the labeling of Figure 4d is confusing. The W799A-D800A+ reads as W799A negative D800A positive.

We understand that this terminology might be confusing for a reader.

For clarity, we replaced in manuscript text and figures:

TALE-Clover by Ctr

TALE-HIRA-Clover by HIRA wt

TALE- HIRA W799A-D800A-Clover by HIRA mut

We simplified the definition of the cells non-expressing the TALE fusion clover negative cells by naming them as non-transfected (NT) and the clover positive cells expressing the TALE fusion as Ctr, HIRA wt and HIRA mut

- In all quantification figures, please state the number of cells used for quantification and not only %.

We added this information in figure legend in the revised version.

- There are several typos in the text and legends, therefore the authors should once again carefully go through the text.

We checked again and hope to have eliminated typos.

Reviewer #3 (Remarks to the Author):

I co-reviewed this manuscript with one of the reviewers who provided the listed reports. This is part of the Nature Communications initiative to facilitate training in peer review and to provide appropriate recognition for Early Career Researchers who co-review manuscripts

We thank this reviewer for her/his evaluation of our work.

References

Camacho, O. V.; Galan, C.; Galán, C.; et al. (2017): Major satellite repeat RNA stabilize heterochromatin retention of Suv39h enzymes by RNA-nucleosome association and RNA:DNA hybrid formation. eLife 6.
<http://dx.doi.org/10.7554/elife.25293>.

Clément, C.; Orsi, G. A.; Gatto, A.; et al. (2018): High-resolution visualization of H3 variants during replication reveals their controlled recycling. Nature Communications 9, 1, 3181. <http://dx.doi.org/10.1038/s41467-018-05697-1>.

Efroni, S.; Duttagupta, R.; Cheng, J.; et al. (2008): Global Transcription in Pluripotent Embryonic Stem Cells. Cell Stem Cell 2, 5, 437-447.
<http://dx.doi.org/10.1016/j.stem.2008.03.021>.

Forest, A.; Quivy, J. P.; Almouzni, G. (2024): Mapping histone variant genomic distribution: Exploiting SNAP-tag labeling to follow the dynamics of incorporation of

H3 variants. *Methods Cell Biol* 182, 49-65.
<http://dx.doi.org/10.1016/bs.mcb.2022.10.007>.

Joron, K.; Viegas, J. O.; Haas-Neill, L.; et al. (2023): Fluorescent protein lifetimes report densities and phases of nuclear condensates during embryonic stem-cell differentiation. *Nat Commun* 14, 1, 4885. <http://dx.doi.org/10.1038/s41467-023-40647-6>.

Lu, J.; Gilbert, D. M. (2007): Proliferation-dependent and cell cycle-regulated transcription of mouse pericentric heterochromatin. *Journal of Cell Biology* 179, 3, 411-421. <http://dx.doi.org/10.1083/jcb.200706176>.

Martens, J. H.; O'Sullivan, R. J.; Braunschweig, U.; et al. (2005): The profile of repeat-associated histone lysine methylation states in the mouse epigenome. *EMBO J* 24, 4, 800-812. <http://dx.doi.org/10.1038/sj.emboj.7600545>.

Martini, E.; Roche, D. M.; Marheineke, K.; et al. (1998): Recruitment of phosphorylated chromatin assembly factor 1 to chromatin after UV irradiation of human cells. *J Cell Biol* 143, 3, 563-575. <http://dx.doi.org/10.1083/jcb.143.3.563>.

Novo, C. L.; Tang, C.; Ahmed, K.; et al. (2016): The pluripotency factor Nanog regulates pericentromeric heterochromatin organization in mouse embryonic stem cells. *Genes & Development* 30, 9, 1101-1115.
<http://dx.doi.org/10.1101/gad.275685.115>.

Novo, C. L.; Wong, E. V.; Hockings, C.; et al. (2022): Satellite repeat transcripts modulate heterochromatin condensates and safeguard chromosome stability in mouse embryonic stem cells. *Nature Communications* 13, 1, 1-16.
<http://dx.doi.org/10.1038/s41467-022-31198-3>.

Percharde, M.; Bulut-Karslioglu, A.; Ramalho-Santos, M. (2017): Hypertranscription in Development, Stem Cells, and Regeneration. *Dev Cell* 40, 1, 9-21.
<http://dx.doi.org/10.1016/j.devcel.2016.11.010>.

Ray-Gallet, D.; Woolfe, A.; Vassias, I.; et al. (2011): Dynamics of Histone H3 Deposition In Vivo Reveal a Nucleosome Gap-Filling Mechanism for H3.3 to Maintain Chromatin Integrity. *Molecular Cell* 44, 6, 928-941.
<http://dx.doi.org/10.1016/j.molcel.2011.12.006>.

Torné, J.; Ray-Gallet, D.; Boyarchuk, E.; et al. (2020): Two HIRA-dependent pathways mediate H3.3 de novo deposition and recycling during transcription. *Nature Structural & Molecular Biology*, 1-12. <http://dx.doi.org/10.1038/s41594-020-0492-7>.

Tosolini, M.; Brochard, V.; Adenot, P.; et al. (2018): Contrasting epigenetic states of heterochromatin in the different types of mouse pluripotent stem cells. *Scientific Reports* 8, 1, 5776. <http://dx.doi.org/10.1038/s41598-018-23822-4>.

Point by point answer to reviewer #1 comments (our response are in blue italics)

The authors have addressed my primary concerns. However, I have a few minor comments that should be addressed to further strengthen the manuscript.

We thank the reviewer for her/his positive evaluation of our revised manuscript

While I now understand the rationale behind selecting HIRA to target H3.3 to the chromocenter, I believe it is important for the authors to explain this reasoning more explicitly in the text.

We added a sentence in the result section to explain the reasoning :

‘ Importantly, for this “forced” deposition of H3.3, we chose HIRA, a chaperone that do not “normally act at the site” in a WT background, rather than overexpressing a histone chaperone already present at PHC such as DAXX/ATRX. The TALE-HIRA fusion fulfilled these requirements to challenge a situation and perturb chromatin dynamics (in this case, H3.1/H3.3 enrichment), a key prerequisite to further develop our imaging analysis pipeline.’

Additionally, the authors may consider discussing whether artificial targeting using the ATRX/DAXX complex (which could recruit H3K9me3/HP1 α) might result in a different outcome, particularly in terms of promoting the reduction of H3.1 deposition and inducing changes in nuclear morphology. A discussion on this connection would enhance the overall conclusions.

It is interesting that wild-type HIRA targeting did not significantly affect major satellite transcription. Does this truly suggest that transcription can be bypassed and that HIRA-mediated H3.3 deposition alone is sufficient to challenge H3.1 deposition at the chromocenter? The authors may consider discussing their findings in the context of observations made in SUVAR3-9-depleted cells, where increased major satellite transcription and enhanced H3.3 deposition have been observed, accompanied by a reduction in H3.1 deposition. This comparison could provide valuable insights into the underlying mechanisms of histone deposition dynamics. Additionally, the enrichment of labile H3.1 and the turnover of H3.1/H3.3 at the chromocenter in ESCs may still be transcription-dependent, a point worth exploring further.

I agree with the authors that the choice of histone chaperones involved in H3.1/H3.3 deposition may be a key determinant of nucleosomal composition. Expanding on this point could provide additional depth to the discussion.

We added in the discussion section sentences to elaborate further based on reviewer’s comments

'Whether artificial targeting using the ATRX/DAXX complex might result in a different outcome would be interesting to explore especially considering its ability to recruit H3K9me3/HP1 α to assess whether it could promote reduced H3.1 deposition and changes in nuclear morphology.'

'Considering that forced HIRA-mediated H3.3 deposition challenged H3.1 deposition at the chromocenter, it is possible that the increased major satellite transcription allows more H3.3 deposition. Thus, it will be important to examine further how the enrichment of labile H3.1 and the turnover of H3.1/H3.3 at the chromocenter in ESCs relates to transcription and its connection with mechanisms of histone deposition.'

Finally, the label on Figure S5d appears to be incorrect and should be corrected
We thank the reviewer for spotting this mistake. We corrected the inversion of CENPA and Clover labels.